# What empowerment indicators are important for food consumption for women? Evidence from 5 sub-Sahara African countries

**Michael Nnachebe Onah**[1,2]*, **Sue Horton**[1], **John Hoddinott**[3]

**1** School of Public Health and Health Systems, Faculty of Applied Health Sciences, University of Waterloo, Ontario, Canada, **2** Institute of Public Policy and Administration, Graduate School of Development, University of Central Asia, Bishkek, Kyrgyzstan, **3** Division of Nutritional Sciences, Cornell Institute of Public Affairs, Cornell University, Ithaca, New York, United States of America

* chebis.onah@gmail.com

## Abstract

This paper draws on data from five sub-Sahara African countries; Uganda, Rwanda, Malawi, Zambia, and Mozambique consisting of 10,041 married women who were cohabitating with a male spouse. The study aim was to investigate the relationship between women's empowerment and women's dietary diversity and consumption of different food items. Women's empowerment was measured using the indicators in the five domains of Women's Empowerment in Agriculture index (WEAI) and women's dietary diversity and food consumption was examined using the women's dietary diversity score (WDDS) measure. OLS and LPM regressions were used and analyses were confirmed using marginal effects from Poisson and logistic regressions. Results suggest that three out of the 10 WEAI indicators of empowerment showed different magnitude and direction in significant associations with improved WDDS and varied associations were found in three out of the five countries examined. In addition, the three significant empowerment indicators were associated with the consumption of different food groups in three out of the five countries examined suggesting that diverse food groups account for the association between the WEAI and WDDS. Improved autonomy, and input in production were associated with improved likelihoods of consumption of dairy products, and fruits and vegetables including vitamin A-rich produce. Empowerment in public speaking was associated with improved consumption of other fruits and vegetables including vitamin A-rich produce. The varied nature of empowerment indicators towards improving women's dietary diversity and food consumption suggests that different empowerment strategies might confer different benefits towards the consumption of different food groups. Further, findings imply that interventions that seek to empower women should tailor their strategies on existing contextual factors that impact on women

## Introduction

Improving women's nutrition outcomes through better access to a more diverse diet has been identified as a key strategy towards improving the lives and livelihoods of women [1–3]. This

**Data Availability Statement:** The data that was utilized for this study can be accessed at the Feed the Future Baseline Data website. For specific countries, the data is open sourced at: Malawi:

https://data.usaid.gov/Agriculture/Feed-The-Future-Malawi-Baseline-Household-Survey/6gc3-3fwr
Mozambique: https://data.usaid.gov/Agriculture/Feed-the-Future-Mozambique-Baseline-Population-Sur/8zqa-4rrz Rwanda: https://data.usaid.gov/Agriculture/Feed-The-Future-Rwanda-Baseline-Household-Survey/8ecv-2gx7 Uganda: https://data.usaid.gov/Agriculture/Feed-The-Future-Uganda-Population-Based-Survey/rbym-kbp6 Zambia: https://data.usaid.gov/Agriculture/Feed-The-Future-Zambia-Baseline-Population-Based-S/4vjz-rqba
These datasets can be downloaded and merged for analyses.

**Funding:** The authors received no specific funding for this work.

**Competing interests:** The authors have declared that no competing interests exist.

could be attributed to the link between improved dietary intake and health outcomes and by extension, enhanced economic productivity [4, 5]. Dietary diversity is usually measured as a snap-shot of women's consumption of different food items with a recall period of 24 hours to 15 days [6]. Many measures of dietary diversity have been developed for specific population groups including the dietary diversity score for women of reproductive age (WDDS) [7]. The WDDS measures women's consumption of nine food groups over a 24-hour recall period and mean scores are used to determine levels of dietary diversity. The WDDS has been used in numerous studies which have examined the relationship between important factors that impact on women's wellbeing and women's dietary diversity [4, 8, 9]. Beyond the performance of individuals on dietary diversity measures, a key area of importance is to further examine which food groups within the dietary diversity measures account for the mean scores obtained and also which factors are most related with the consumption of these food groups.

A key factor that has been identified to improve women's well-being along with food consumption is empowerment, and many studies have applied different measures of women's empowerment to examine this relationship [10–16]. These measures vary in the domains and indicators of empowerment, and construction and focus. For instance, while some measures focus on women in rural agrarian settings [17, 18], others focus on women in sub-Saharan Africa [19] or more narrowly, on women in east Africa [14] hence, there is no consensus on how empowerment should be operationalised and measured [20]. This makes the applicability and comparability of empowerment measures across different settings difficult. However, there is consensus that empowerment is a complex, relational, and multidimensional concept where different domains and indicators are relevant for specific outcomes including women's nutrition [14, 21–26]. Studies have shown a strong link between improved women's empowerment in access and control of income and assets [21], aggregate empowerment score [27, 28], autonomy in production and work hours [29], agricultural decisions [30], and credit decisions [31] on women's food consumption and nutrition outcomes. Further, women's empowerment in nutrition has also been found to influence positive nutrition outcomes including body mass index [32] and anemia [33].

The generalizability of these study findings is however limited since women's empowerment is heavily influenced by prevailing contextual norms and factors [23, 34]. In many LMICs especially in sub-Saharan Africa (SSA), existing cultural norms and practices tend to be patriarchal and patrilineal which favour men, and these norms further create a gender divide that limit women's empowerment and access to resources including food [30, 35, 36]. While these norms are prevalent in many LMICs, they tend to be context-specific and vary within and across countries and hence, have differential impact on women. This can also be observed in the varied nature of associations between women's empowerment domains and dietary diversity and nutrition outcomes [31, 37–39]. Further, while contextual factors including norms, myths, and taboos influence women's ability to gain and express empowerment, these factors also influence women's food consumption [40–42]. One of the well-known measures of women's empowerment is the Women's Empowerment in Agriculture Index (WEAI) [17]. Since its development, the measure has been used in numerous studies (predominantly in Asia) with varied findings on the role of different empowerment indicators and women's dietary diversity and nutrition outcomes [31, 43–49]. The varied nature of the relationship between the indicators of empowerment in the WEAI and women's food consumption and nutrition outcomes also suggest that the extent to which women are able to express empowerment is context specific. Prevailing contextual factors and norms dictate women's role and affect how women's empowerment changes their food intake [10, 31, 50].

In the context of improving women's access to diverse foods through empowerment, the need to examine which empowerment domains are relevant for the consumption of specific

food items in dietary diversity measures are important. The importance is further highlighted since women use a mix of food production and purchase in food consumption [37]. Hence, different empowerment domains might have different effects on not only women's dietary diversity but also on the consumption of different food groups within dietary diversity measures. This is an area that is missing in literature for which the present study aims to examine. We undertook this study in order to better understand how women's empowerment indicators impacts on women's diets and dietary diversity in different countries. Analyses were conducted across five SSA countries (Mozambique, Rwanda, Malawi, Uganda, and Zambia).

## Data, empirical specification and variables

The present study was conducted using secondary data from the Feed the Future Baseline Survey, for which the ethics approvals for each country was obtained prior to surveys. The data is de-identified and hence the analyses presented here do not require additional human subjects review and ethics approval.

### The Feed the Future study (FTF)

The Feed the Future programme is a United States Government initiative that seeks to address global food insecurity by focusing on growth of the agricultural sector and improvement in nutritional status in 19 developing countries. The United States Agency for International Development (USAID) is responsible for leading the government-wide effort to implement the Feed the Future initiative. The programme seeks to reduce poverty and undernutrition in 19 developing countries by focusing on accelerating growth of the agricultural sector, addressing root causes of undernutrition, and reducing gender inequality. The main target of the programme is "to reduce by 20% the prevalence of poverty and the prevalence of stunted children under five years of age in the areas where we (USAID personnel) work" [51].

Data was extracted from five countries in SSA out of the 19 countries where the programme has been implemented and a baseline survey was conducted between 2010–2013. These were low-and-lower-middle income countries in east (Uganda and Rwanda) and southern (Zambia, Malawi, and Mozambique) Africa with GDP (PPP) per capita ranging from as low as USD1,172 in Malawi to USD3,997 in Zambia. There were also wide ranges of wealth inequalities in these countries with as much as 61% of Zambians living below the domestic poverty line as of 2012 and 20% of Ugandans living below the poverty line as of 2013 [52].

The Feed the Future study used population-based surveys to collect data. The present study used a total of 10,041 households (1,558 in Zambia, 2,316 in Uganda, 2,856 in Malawi, 2,425 in Mozambique, and 1,831 in Rwanda) where married women in rural settings were interviewed (see Fig 1 for the breakdown of the analysed sample obtained from the FTF).

As part of the modules in the questionnaire used, the WEAI was administered to both the male and female decision-makers in surveyed households. As indicated above, our study has been restricted to sampled households in rural areas with women in a union and with information on women's dietary diversity where the relationship between women's empowerment as measured by the WEAI and women's consumption of different food groups (dietary diversity) was explored.

### The Women's Empowerment in Agriculture Index (WEAI)

Alkire et al. [17] developed the WEAI based on research evidence on agency and empowerment, including the works of Alsop, Bertelsen, and Holland [53], Ibrahim and Alkire [25], Narayan-Parker [54], and Narayan, Pritchett, and Kapoor [55]. This research evidence proposed domain-specific measures of empowerment constructed from questions that could be

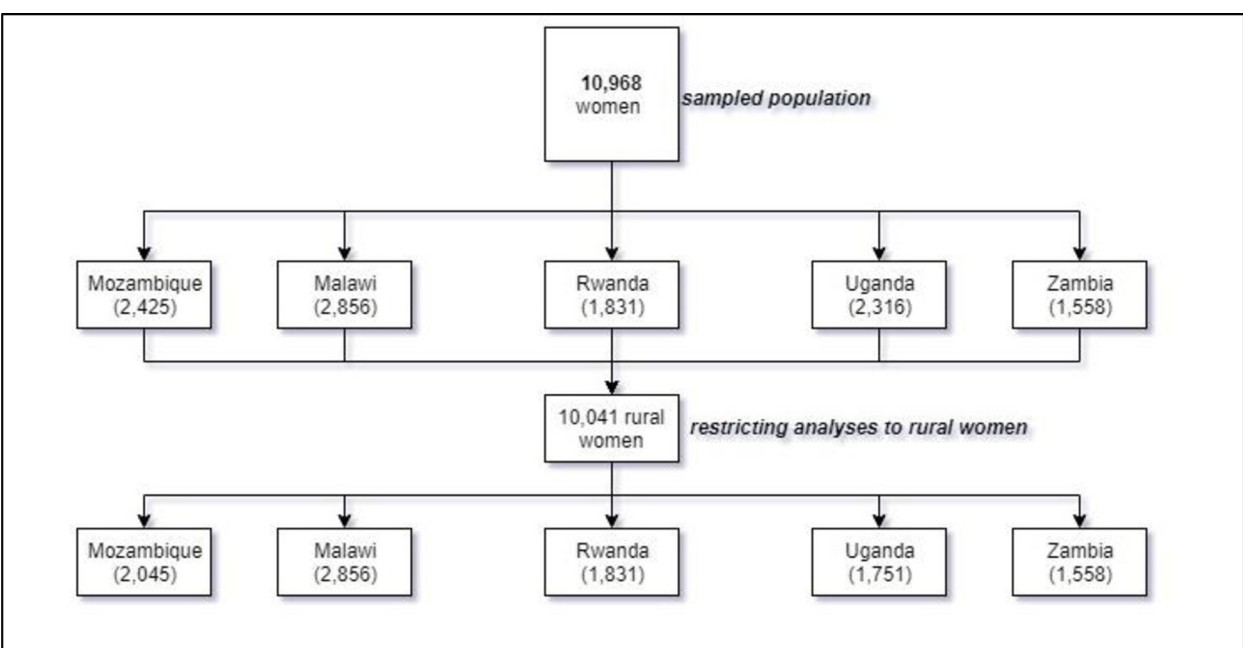

**Fig 1. Breakdown of FTF sample.**

administered in individual or household surveys. Based on the methodology reported in Alkire and Foster [56], the WEAI is an aggregate index, reported at the country or regional level, based on individual-level data collected by interviewing men and women within the same households. Ten indicators of the WEAI reflect the percentage of women who are empowered in five domains of empowerment (5DE) in agriculture. These five domains of the WEAI are (1) decisions about agricultural production, (2) access to and decision-making power about productive resources, (3) control of use of income, (4) leadership in the community, and (5) time allocation based on a log of different activities, each of the five domains receives equal weight. Table 1 presents the domains, indicators, and weights in the WEAI. The responses for the questions contained in the index include both dichotomous (Yes/No) and categorical options, where up to 10 different options were offered (for more information, see Alkire et al., [17]).

Across the five SSA countries, disempowerment in the resources domain contributed the most (approx. 30% - 38%) towards women's disempowerment (see Fig 2). This was followed by disempowerment in the leadership domain (approx. 20% - 25%), while disempowerment in production decisions contributed the least (approx. 9% - 13%) towards women's disempowerment. Further disaggregated analysis of the contribution of the 10 sub-domains towards women's disempowerment can be found in Fig 2.

## Women's dietary diversity outcome variable

The women's dietary diversity score (WDDS) is an indicator used to examine the quality of diet for women of reproductive age [7, 57]. The indicator is recommended by the United Nations Food and Agriculture Organisation (FAO) as a simple and valid measure of women's dietary diversity through the consumption of food items belonging to nine food groups. The WDDS is measured as the number of food groups consumed by women 15–49 years in the past 24 hours where the food groups include; (1) Starchy foods; (2) Legumes and nuts; (3)

**Table 1. Domains, indicators, and weights in the WEAI.**

| Domain | Indicator | Definition of indicator | Weight |
|---|---|---|---|
| Production | 1.1 Input in productive decisions | Sole or joint decision-making over food and cash-crop farming, livestock, and fisheries | 1/10 |
|  | 1.2 Autonomy in production | Autonomy in agricultural production reflects the extent to which the respondent's motivation for decision-making reflects own values rather than a desire to please others or avoid harm | 1/10 |
| Resources | 2.1 Ownership of assets | Sole or joint ownership of major household assets | 1/15 |
|  | 2.2 Purchase, sale, or transfer of assets | Whether respondent participates in decision to buy, sell, or transfer assets | 1/15 |
|  | 2.3 Access to and decisions about credit | Access to and participation in decision-making concerning credit | 1/15 |
| Income | 3.1 Control over use of income | Sole or joint control over income and expenditures | 1/5 |
| Leadership | 4.1 Group membership | Whether respondent is an active member in at least one economic or social group | 1/10 |
|  | 4.2 Speaking in public | Whether the respondent is comfortable speaking in public concerning issues relevant to oneself or one's community | 1/10 |
| Time | 5.1 Workload | Allocation of time to productive and domestic tasks | 1/10 |
|  | 5.2 Leisure | Satisfaction with time for leisure activities | 1/10 |

Source:(Ewerling et al. 2017 [19]).

Dairy products; (4) Flesh foods; (5) Eggs; (6) Dark-green leafy vegetables; (7) Vitamin-A rich fruits and vegetables; (8) Other fruits and vegetables; and (9) Organ meat. According to the FAO guidelines, mean scores determine performance on the WDDS, that is, women that had higher mean scores experienced better dietary diversity. A variant of the WDDS has been created, the Minimum Dietary Diversity for Women (MDD-W) measure, which separates legumes, nuts, and seeds into two food groups; legumes, and nuts and seeds, and subsumes organ meat into animal meat food group [58], however the present study has used the original WDDS due to data limitations.

The FTF data for the five SSA countries contains information on the women's consumption of nine food groups and WDDS was examined as a continuous variable. In addition, individual food groups were examined as dichotomous outcome variables in line with our hypothesis that the different domains and indicators of women's empowerment would have differential impacts on women's consumption of individual food groups. This was also informed by the qualitative study on economic empowerment and women's dietary diversity conducted by the same researchers [59] where we found that economic empowerment would be important for the consumption of some food items including expensive-to-purchase items but would not be as important for the consumption of staple food items.

## Key independent variables

The WEAI index aggregate score and 10 empowerment indicators contained in the five domains of empowerment were used as the key independent variables and these domains are described in Table 1. The WEAI was examined as an aggregate score in one regression model and each of the empowerment indicators were examined in 10 separate regression models. Our hypothesis was that the aggregate empowerment score and each indicator might have different associations with WDDS and consumption of different food groups for women. The results presented here first include results for the aggregate score and each of the 10 WEAI indicators and subsequently focus on the indicators that show a significant association with WDDS.

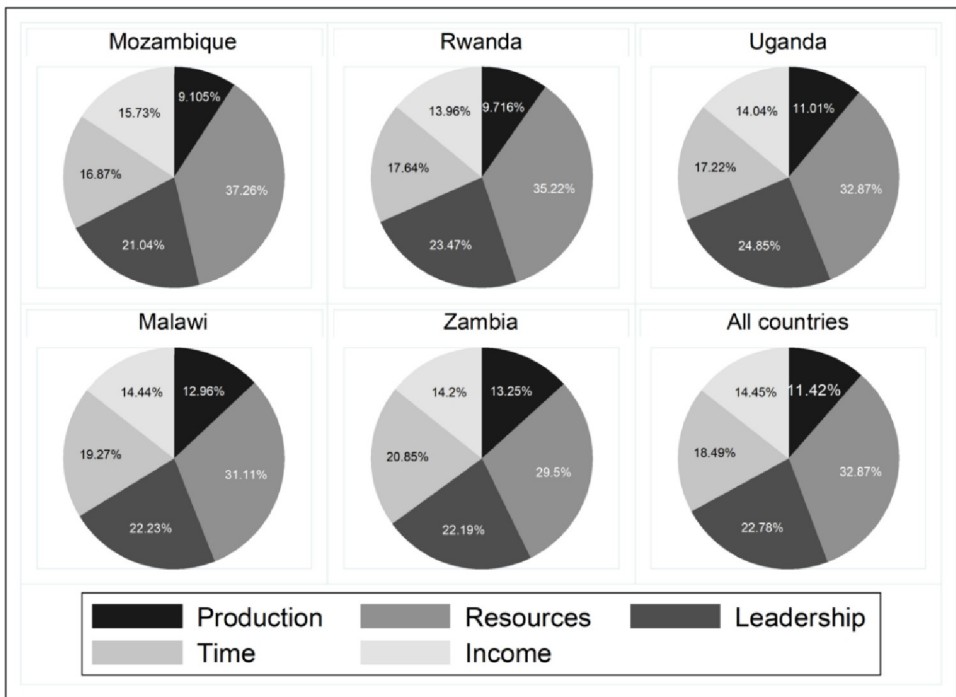

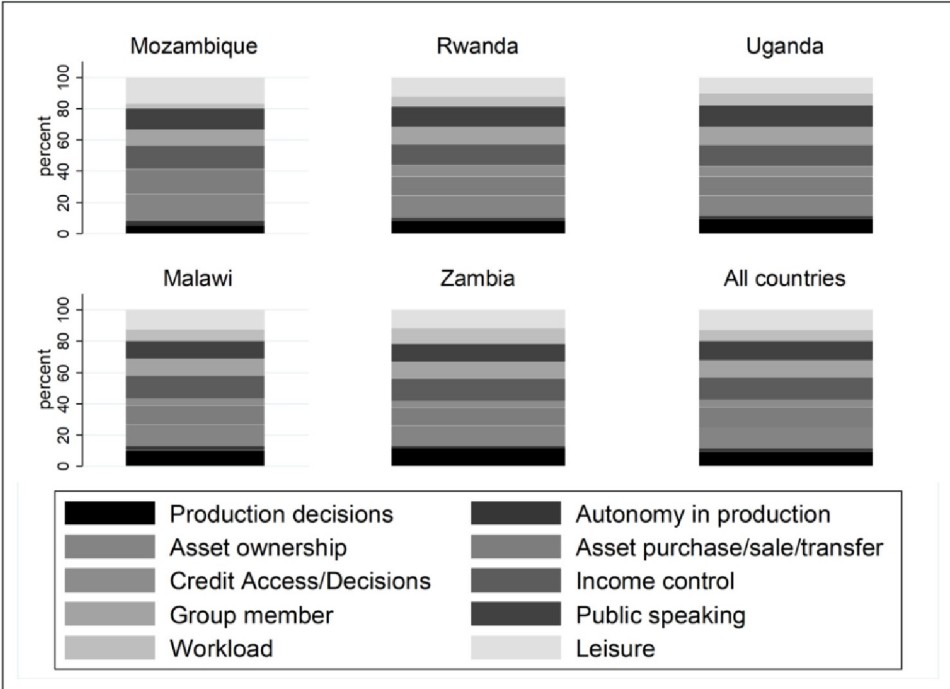

**Fig 2. Contribution of the five domains and 10 indicators to women's disempowerment.**

## Control variables

A limitation of the present study is the lack of an exhaustive list of all relevant control variables that might be associated with women's nutrition. We limited the control variables to those available in the study data and included women's and men's sociodemographic characteristics

and household SES index. Other survey characteristics including month of study and study districts were also included to control for spatial and seasonal factors. In addition, a dummy variable for individual countries were included in the pooled analyses to control for the effect of individual countries on the outcomes of interest.

**Age.** Women's and men's age in years were included as control variables as empowerment is expected to increase with age, as with dietary diversity [31].

**Education.** Women's education in years was estimated as a control variable based on the hypothesis that improved educational attainment for women is linked with improved knowledge transition within households and socioeconomic status (SES) of household members, and that this could influence women's ability to consume more diverse diets [60, 61].

**Household size.** Household size was estimated as a continuous control variable based on the hypothesis that in rural settings such as in the present study, income-generation potentials are limited, and household size would be an important determinant of household's SES and individuals' ability to achieve improved dietary diversity [62].

**Socioeconomic status index.** We constructed this socioeconomic status (SES) index using a principal component analysis (PCA). The PCA used a list of household dwelling characteristics and living conditions to estimate SES [63, 64]. These variables reflect some variations across household living conditions and were pooled together to construct a SES index. The variables include the type of house roofing, floor, and external walls, source of drinking water and fuel, and availability of electricity. We included only variables with moderate to high correlation coefficients between them in building the index (S18 Table). The assumption was that women belonging to higher SES households would perform better on the WDDS. To control for possible nonlinear relationship between the SES index and WDDS, we included the index and index-squared.

**Study districts.** This dummy control variable detail the number of districts across the five SSA countries that were included in the FTF baseline studies and was included based on the assumption that different districts would have different economic and agricultural activities, and geographical characteristics including rainfall patterns and drought which could affect food availability and influence women's food consumption. In total, 38 study districts (Mozambique; 22, Malawi, 7, Rwanda; 27, Uganda; 37, and Zambia; 6) were included in the pooled and disaggregated analyses.

**Study month.** This dummy control variable provides information on which months of the year the study was conducted and was included based on the assumption that timing of study would have an important effect on individuals' and households' food consumption. This is because there are important seasonal variations in food availability and timing of study might reflect these variations [65].

## Empirical specification

To specify the model, let $Y_i$ be the outcome variable (WDDS) treated as a continuous variable using the OLS model and estimated as:

$$Y_i = \beta_0 + \beta_1 WEAI + \beta_2 C + \beta_3 I + \varepsilon \qquad 1$$

where C denotes the control variables; $I$ stands for other demographic covariates; $\beta_1, \beta_2, and \beta_3$ are the estimated parameters/parameter vectors; and $\varepsilon$ is the error term. The WEAI index has been treated as an aggregate empowerment score in one model, and other 10 sub-domains were included in different models based on our hypothesis.

Estimating the different food groups as contained in the WDDS, we have used linear probability models (LPMs). Let $y_i$ be the outcome variable (different food groups in the WDDS)

estimated as:

$$y_i = \sigma_0 + \sigma_1 WEAI + \sigma_2 C + \sigma_3 I + \varepsilon \qquad 2$$

where C stands for the control variables; $I$ is for other demographic covariates; $\sigma_1$, $\sigma_2$, and $\sigma_3$ are the parameters to be estimated and interpreted as the change in the probability that $y_i = 1$, holding constant the other regressors, and $\varepsilon$ is the error term. The WEAI has been treated the same as described in Eq 1.

Stata version 15.1 was used to perform multiple regression analysis to explore the relationships between identified dependent and independent variables. Descriptive statistics were used to summarise sample statistics including the 10 WEAI indicators. The OLS regression model, adjusting for the effects of specified covariates, has been used to examine significant associations between women's empowerment and WDDS. LPMs were used, also adjusting for covariates, to examine the relationship between significant domains in the OLS for WDDS and the consumption of different food groups, and analyses also focused on countries with significant associations in the OLS. For these analyses, cluster sampling effects (including sampling bias and the lack of independent and identically distributed properties) have been controlled for and appropriate sampling weights have been applied. To test the correctness of the regression approaches used, marginal effects from Poisson and logistic regression models were used to test the OLS and LPM models. For more details, see [37]. Significance was established at 95% and 99% confidence intervals.

## Findings

### Sample characteristics

On average and across the five SSA countries examined, women consumed three out of nine food groups of the WDDS except for Zambia where women consumed food items belonging to four food groups (Table 2). On average, household size was six and across countries, Malawi recorded the highest prevalence of hunger based on a mean hunger score of 1.14 while Mozambique recorded the lowest score of 0.65 on average. The average age for women was 28.95 years (31.85 years for men) and women on average had 5.50 years of schooling with Uganda recording the highest years of schooling. For variables included in the SES index, the most common type of roof material was thatch (52%) followed by corrugated metal (24.7%) and nearly all households used firewood as fuel (92.5%). Fifty-six percent of the variation in

**Table 2. Sample characteristics (percentages, except where otherwise specified).**

|  | All | Mozambique | Malawi | Rwanda | Uganda | Zambia |
|---|---|---|---|---|---|---|
| *Outcome variable* |  |  |  |  |  |  |
| 1. Sum of food groups consumed (out of 9) | 3.38 (SD: 1.31) | 3.39 (SD: 1.40) | 3.49 (SD: 1.27) | 3.43 (SD: 1.34) | 3.36 (SD: 1.29) | 4.10 (SD: 1.13) |
| *Women and household characteristics* |  |  |  |  |  |  |
| 2. Women's age in years (mean) | 28.95 (SD: 9.30) | 29.04 (SD: 9.57) | 28.36 (SD: 8.92) | 29.63 (SD: 9.67) | 28.44 (SD: 9.61) | 29.26 (SD: 9.74) |
| 3. Men's age in years (mean) | 31.85 (SD: 12.87) | 32.05 (SD: 12.87) | 31.41 (SD: 14.12) | 32.78 (SD:14.43) | 31.42 (SD: 14.59) | 31.68 (SD: 14.80) |
| 4. Women's education in years (mean) | 5.50 (SD: 4.28) | 2.22 (SD: 0.88) | 2.45 (SD: 0.98) | 2.99 (SD: 1.01) | 7.92 (SD: 4.62) | 5.12 (SD: 2.90) |
| 5. Household hunger scale [0–6] (mean) | 0.97 (SD: 1.31) | 0.65 (SD: 1.68) | 1.14 (SD: 1.33) | 1.34 (SD: 1.30) | 0.79 (SD: 1.24) | 0.69 (SD: 1.12) |
| 6. Household size | 6.09 (SD: 2.53) | 5.32 (SD: 2.51) | 5.29 (SD: 1.96) | 5.30 (SD: 1.90) | 6.98 (SD: 2.75) | 6.69 (SD: 2.99) |
| 7. SES index: *Q1* | 26.19 | 23.19 | 23.51 | 25.31 | 30.54 | 34.94 |
| *Q2* | 28.86 | 26.38 | 25.76 | 25.98 | 25.19 | 34.51 |
| *Q3* | 22.98 | 25.56 | 22.83 | 23.40 | 21.67 | 21.92 |
| *Q4* | 21.97 | 20.76 | 18.67 | 19.62 | 16.63 | 7.25 |

the socioeconomic index created by the principal component analysis was explained by the first two component factors (household roofing and floor type) while the least variation was explained by the last two component factors (electricity and fuel source). The heteroskedastic bootstrap confidence intervals were also narrow, and results from the correlation matrix indicate that there was no significant correlation among the variables included in the SES index (for more details, see [37]).

## Summary statistics of the 10 WEAI indicators

Findings suggest that there were important variations in women's performance across 4 out of the 10 WEAI indicators with further variations across the five SSA countries examined (Table 3). These variations were found in: (1) two production domain indicators (*autonomy and input in production domains and activities*); (2) one resources domain indicator (*sole/joint decision-making regarding credit and credit source*): and (3) one leisure domain indicator (*non-excessive in workload*). This implies that a majority (in most cases, over three-quarters) of women met the thresholds and were considered empowered in six out of the 10 empowerment indicators as defined by Alkire et al. [17].

## Women's empowerment and dietary diversity

To further test which empowerment indicators exhibit important associations with women's dietary diversity, regression models were specified, and coefficients presented in Tables 4–14 (see S1–S11 Tables for confirmatory regression analyses using marginal effects of Poisson regression). Findings suggest that the two indicators in the production domain (autonomy in production decision, and input in production decisions and activities) and one leadership domain indicator (comfortable speaking in public) exhibited significant positive associations with improved WDDS. These findings were also confirmed in the Poisson regression models with marginal effects where coefficients were broadly similar to those from the OLS regression models.

**Table 3. Descriptive statistics of the 10 WEAI indicators.**

|  | All | Mozambique | Malawi | Rwanda | Uganda | Zambia |
|---|---|---|---|---|---|---|
| *Empowerment indicators* |  |  |  |  |  |  |
| Empowerment score, = 1 if empowered | 0.81 (SD: 0.13) | 0.80 (SD: 0.14) | 0.79 (SD: 0.14) | 0.84 (SD: 0.12) | 0.81 (SD: 0.13) | 0.77 (SD: 0.13) |
| *Production domain* |  |  |  |  |  |  |
| 1. Autonomy in ≥ 1 activity linked to production | 0.14 (SD: 0.34) | 0.17 (SD: 0.37) | 0.23 (SD: 0.42) | 0.12 (SD: 0.32) | 0.12 (SD: 0.32) | 0.11 (SD: 0.31) |
| 2. Input in ≥ 2 domains of production | 0.66 (SD: 0.47) | 0.33 (SD: 0.47) | 0.71 (SD: 0.45) | 0.65 (SD: 0.48) | 0.70 (SD: 0.46) | 0.83 (SD: 0.37) |
| *Resources domain* |  |  |  |  |  |  |
| 3. Sole/joint ownership of ≥ 2 small or 1 large asset | 0.93 (SD: 0.24) | 0.94 (SD: 0.24) | 0.93 (SD: 0.25) | 0.97 (SD: 0.16) | 0.92 (SD: 0.27) | 0.90 (SD: 0.30) |
| 4. Sole/joint right over ≥ 1 type of asset transaction | 0.85 (SD: 0.34) | 0.91 (SD: 0.21) | 0.83 (SD: 0.36) | 0.87 (SD: 0.33) | 0.85 (SD: 0.35) | 0.83 (SD: 0.38) |
| 5. Sole/joint decision-making regarding ≥ 1 credit source | 0.47 (SD: 0.50) | 0.12 (SD: 0.32) | 0.37 (SD: 0.48) | 0.57 (SD: 0.50) | 0.51 (SD: 0.50) | 0.39 (SD: 0.47) |
| *Income domain* |  |  |  |  |  |  |
| 6. Input in decisions on wage, employment, minor expenses | 0.93 (SD: 0.24) | 0.83 (SD: 0.37) | 0.95 (SD: 0.22) | 0.95 (SD: 0.22) | 0.95 (SD: 0.22) | 0.96 (SD: 0.20) |
| *Leadership domain* |  |  |  |  |  |  |
| 7. Membership of ≥ 1 group | 0.81 (SD: 0.31) | 0.66 (SD: 0.47) | 0.78 (SD: 0.42) | 0.83 (SD: 0.37) | 0.84 (SD: 0.37) | 0.74 (SD: 0.44) |
| 8. Comfortable speaking in public in ≥ 1 context | 0.87 (SD: 0.33) | 0.79 (SD: 0.41) | 0.78 (0.41) | 0.87 (SD: 0.33) | 0.91 (SD: 0.28) | 0.84 (SD: 0.37) |
| *Leisure domain* |  |  |  |  |  |  |
| 9. Avg workload >10.5hrs in 24hrs (excessive workload) | 0.52 (SD: 0.49) | 0.19 (SD: 0.39) | 0.54 (SD: 0.59) | 0.53 (SD: 0.49) | 0.55 (SD: 0.49) | 0.74 (SD: 0.43) |
| 10. Satisfaction with leisure time | 0.80 (SD: 0.40) | 0.94 (SD: 0.24) | 0.84 (SD: 0.37) | 0.87 (SD: 0.34) | 0.73 (SD: 0.45) | 0.83 (SD: 0.38) |

**Table 4. OLS regression for WDDS–empowerment score.**

|  | (1) | (2) | (3) | (4) | (5) | (6) |
|---|---|---|---|---|---|---|
|  | **Pooled** | **Mozambique** | **Rwanda** | **Malawi** | **Uganda** | **Zambia** |
| Empower score | 0.095 | 0.63** | 1.304*** | -0.008 | -0.485 | -0.084 |
|  | (0.203) | (0.332) | (0.356) | (0.22) | (0.358) | (0.259) |
| SES index | -0.047 | 0.034 | 0.569 | -0.269** | -0.932** | -10.745** |
|  | (0.108) | (0.333) | (10.093) | (0.124) | (0.438) | (0.678) |
| SES index squared | 0.02 | 0.156 | 0.179 | 0.021 | 0.148** | -0.986** |
|  | (0.015) | (0.218) | (0.357) | (0.014) | (0.06) | (0.41) |
| Men's age | 0.005*** | 0.007* | 0.002 | 0.006* | 0.008*** | 0.003 |
|  | (0.002) | (0.004) | (0.002) | (0.003) | (0.003) | (0.003) |
| Women's age | -0.012*** | -0.009** | -0.01*** | -0.016*** | -0.013*** | -0.003 |
|  | (0.002) | (0.004) | (0.004) | (0.003) | (0.004) | (0.003) |
| Women's education | 0.039*** | 0.066 | 0.105*** | 0.085** | 0.031*** | 0.04*** |
|  | (0.009) | (0.063) | (0.035) | (0.04) | (0.01) | (0.013) |
| Household size | 0.041*** | 0.049* | 0.045 | 0.045* | 0.025 | 0.046*** |
|  | (0.012) | (0.027) | (0.034) | (0.023) | (0.017) | (0.013) |
| Study location | -0.014** | 0.076*** | 0.022*** | 0.027 | -0.028*** | -0.078 |
|  | (0.005) | (0.016) | (0.008) | (0.057) | (0.007) | (0.073) |
| Study month[a] |  |  |  |  |  |  |
| February | 0.117 | 0.004 |  |  |  |  |
|  | (0.245) | (0.132) |  |  |  |  |
| March | -0.614*** | -0.416** |  |  |  |  |
|  | (0.205) | (0.186) |  |  |  |  |
| April | -0.185 | 0.368 |  |  |  |  |
|  | (0.243) | (0.268) |  |  |  |  |
| November | -0.03 | 0.287** |  | -2.298*** | 0.36 |  |
|  | (0.165) | (0.135) |  | (0.121) | (0.28) |  |
| December | 0.139 | -0.448*** | 0.242** | -2.163*** | -0.221 | -0.043 |
|  | (0.127) | (0.165) | (0.112) | (0.269) | (0.201) | (0.207) |
| Countries [*Ref: Mozambique*] |  |  |  |  |  |  |
| Malawi | -0.209 |  |  |  |  |  |
|  | (0.203) |  |  |  |  |  |
| Rwanda | -0.289* |  |  |  |  |  |
|  | (0.17) |  |  |  |  |  |
| Uganda | -0.795** |  |  |  |  |  |
|  | (0.37) |  |  |  |  |  |
| Zambia | 0.012 |  |  |  |  |  |
|  | (0.165) |  |  |  |  |  |
| Observations | 18117 | 2100 | 3681 | 4569 | 3754 | 4013 |
| R-squared | 0.053 | 0.153 | 0.058 | 0.037 | 0.093 | 0.061 |

Note: Standard errors in parentheses

*** $p < 0.01$

** $p < 0.05$

* $p < 0.1$

[a]Ref categories; January (Pooled, Mozambique, Rwanda, Malawi, Uganda), November (Zambia).

**Table 5. OLS regression for WDDS–prod domain (Aut in $\geq$2 production activities).**

| | (1) | (2) | (3) | (4) | (5) | (6) |
|---|---|---|---|---|---|---|
| | **Pooled** | **Mozambique** | **Rwanda** | **Malawi** | **Uganda** | **Zambia** |
| Aut in prod decs | 0.186** | 0.053 | 0.108 | 0.082 | 0.348** | 0.108 |
| | (0.074) | (0.131) | (0.113) | (0.067) | (0.156) | (0.09) |
| SES index | -0.021 | 0.012 | 0.673 | -0.288** | -0.671 | -10.645** |
| | (0.108) | (0.315) | (10.043) | (0.128) | (0.562) | (0.642) |
| SES index squared | 0.019 | 0.14 | 0.209 | 0.022 | 0.115 | -0.915** |
| | (0.015) | (0.217) | (0.342) | (0.014) | (0.075) | (0.393) |
| Men's age | 0.005*** | 0.007** | 0.003 | 0.006* | 0.008*** | 0.003 |
| | (0.001) | (0.004) | (0.002) | (0.003) | (0.003) | (0.003) |
| Women's age | -0.011*** | -0.012*** | -0.01*** | -0.015*** | -0.012*** | -0.003 |
| | (0.002) | (0.004) | (0.004) | (0.003) | (0.003) | (0.003) |
| Women's education | 0.04*** | 0.012 | 0.124*** | 0.088** | 0.03*** | 0.041*** |
| | (0.009) | (0.06) | (0.033) | (0.039) | (0.01) | (0.013) |
| Household size | 0.033** | 0.05* | 0.05 | 0.036* | 0.014 | 0.044*** |
| | (0.013) | (0.027) | (0.032) | (0.021) | (0.019) | (0.013) |
| Study location | -0.014*** | 0.061*** | 0.018** | 0.021 | -0.028*** | -0.069 |
| | (0.005) | (0.013) | (0.008) | (0.055) | (0.006) | (0.071) |
| Study month[a] | | | | | | |
| February | 0.092 | 0.037 | | | | |
| | (0.245) | (0.146) | | | | |
| March | -0.658*** | -0.382** | | | | |
| | (0.205) | (0.188) | | | | |
| April | -0.172 | 0.415* | | | | |
| | (0.231) | (0.249) | | | | |
| November | 0.017 | 0.357** | | -2.294*** | 0.349 | |
| | (0.161) | (0.142) | | (0.123) | (0.293) | |
| December | 0.189 | -0.411** | 0.301*** | -2.157*** | -0.181 | -0.052 |
| | (0.122) | (0.174) | (0.112) | (0.268) | (0.224) | (0.212) |
| Countries [*Ref: Mozambique*] | | | | | | |
| Malawi | -0.19 | | | | | |
| | (0.208) | | | | | |
| Rwanda | -0.272 | | | | | |
| | (0.17) | | | | | |
| Uganda | -0.779** | | | | | |
| | (0.365) | | | | | |
| Zambia | 0.044 | | | | | |
| | (0.164) | | | | | |
| Observations | 19668 | 2594 | 4011 | 4773 | 4063 | 4227 |
| R-squared | 0.052 | 0.12 | 0.046 | 0.036 | 0.09 | 0.06 |

Note: Standard errors in parentheses

*** $p<0.01$

** $p<0.05$

* $p<0.1$

[a]Ref categories; January (Pooled, Mozambique, Rwanda, Malawi, Uganda), November (Zambia).

**Table 6. OLS regression for WDDS–prod domain (Input in $\geq$ 2 productive decisions).**

| | (1) | (2) | (3) | (4) | (5) | (6) |
|---|---|---|---|---|---|---|
| | **All** | **Mozambique** | **Rwanda** | **Malawi** | **Uganda** | **Zambia** |
| Input in productive decs | 0.219*** | -0.071 | 0.525*** | 0.05 | 0.135 | 0.179* |
| | (0.056) | (0.083) | (0.098) | (0.08) | (0.095) | (0.098) |
| SES index | -0.02 | 0.023 | 0.869 | -0.276** | -0.703 | -10.576** |
| | (0.109) | (0.318) | (10.037) | (0.127) | (0.561) | (0.624) |
| SES index squared | 0.019 | 0.154 | 0.268 | 0.021 | 0.12 | -0.863** |
| | (0.015) | (0.219) | (0.341) | (0.015) | (0.075) | (0.379) |
| Men's age | 0.005*** | 0.006* | 0.002 | 0.006* | 0.008*** | 0.003 |
| | (0.002) | (0.004) | (0.002) | (0.003) | (0.003) | (0.003) |
| Women's age | -0.012*** | -0.011*** | -0.01*** | -0.016*** | -0.013*** | -0.003 |
| | (0.002) | (0.004) | (0.004) | (0.003) | (0.004) | (0.003) |
| Women's education | 0.036*** | -0.008 | 0.112*** | 0.083** | 0.027*** | 0.039*** |
| | (0.009) | (0.068) | (0.035) | (0.039) | (0.01) | (0.012) |
| Household size | 0.03** | 0.054** | 0.032 | 0.035* | 0.012 | 0.046*** |
| | (0.014) | (0.027) | (0.03) | (0.021) | (0.02) | (0.013) |
| Study location | -0.015*** | 0.062*** | 0.021** | 0.019 | -0.029*** | -0.07 |
| | (0.005) | (0.014) | (0.008) | (0.056) | (0.007) | (0.07) |
| Study month[a] | | | | | | |
| February | 0.116 | 0.039 | | | | |
| | (0.245) | (0.127) | | | | |
| March | -0.633*** | -0.385** | | | | |
| | (0.207) | (0.187) | | | | |
| April | -0.168 | 0.397 | | | | |
| | (0.232) | (0.257) | | | | |
| November | 0.021 | 0.347** | | -2.288*** | 0.344 | |
| | (0.162) | (0.132) | | (0.122) | (0.313) | |
| December | 0.185 | -0.43*** | 0.306*** | -2.147*** | -0.145 | -0.074 |
| | (0.124) | (0.161) | (0.112) | (0.268) | (0.254) | (0.205) |
| Countries [*Ref: Mozambique*] | | | | | | |
| Malawi | -0.182 | | | | | |
| | (0.208) | | | | | |
| Rwanda | -0.241 | | | | | |
| | (0.169) | | | | | |
| Uganda | -0.781** | | | | | |
| | (0.364) | | | | | |
| Zambia | 0.058 | | | | | |
| | (0.163) | | | | | |
| Observations | 19709 | 2594 | 4031 | 4777 | 4068 | 4239 |
| R-squared | 0.05 | 0.123 | 0.048 | 0.036 | 0.083 | 0.062 |

Note: Standard errors in parentheses

*** p<0.01

** p<0.05

* p<0.1

[a]Ref categories; January (Pooled, Mozambique, Rwanda, Malawi, Uganda), November (Zambia).

**Table 7.  OLS regression results for WDDS–resources domains (sole/joint ownership of $\geq$ 2 small/1 large assets).**

| | (1) | (2) | (3) | (4) | (5) | (6) |
|---|---|---|---|---|---|---|
| | **All** | **Mozambique** | **Rwanda** | **Malawi** | **Uganda** | **Zambia** |
| Sole/Joint asset ownership | 0.048 | 0.319** | 0.449*** | -0.104 | -0.152 | -0.122 |
| | (0.075) | (0.125) | (0.114) | (0.112) | (0.132) | (0.095) |
| SES index | -0.018 | 0.006 | 0.734 | -0.287** | -0.677 | -10.677** |
| | (0.108) | (0.306) | (10.049) | (0.128) | (0.565) | (0.641) |
| SES index squared | 0.019 | 0.131 | 0.227 | 0.023 | 0.116 | -0.944** |
| | (0.015) | (0.209) | (0.343) | (0.015) | (0.076) | (0.391) |
| Men's age | 0.006*** | 0.007* | 0.002 | 0.006* | 0.009*** | 0.003 |
| | (0.002) | (0.004) | (0.002) | (0.003) | (0.003) | (0.003) |
| Women's age | -0.011*** | -0.013*** | -0.01*** | -0.016*** | -0.012*** | -0.003 |
| | (0.002) | (0.004) | (0.004) | (0.003) | (0.003) | (0.003) |
| Women's education | 0.04*** | 0.02 | 0.128*** | 0.088** | 0.031*** | 0.041*** |
| | (0.009) | (0.06) | (0.034) | (0.038) | (0.01) | (0.013) |
| Household size | 0.033** | 0.048* | 0.041 | 0.037* | 0.014 | 0.046*** |
| | (0.013) | (0.027) | (0.031) | (0.02) | (0.019) | (0.013) |
| Study location | -0.014*** | 0.061*** | 0.018** | 0.021 | -0.028*** | -0.073 |
| | (0.005) | (0.013) | (0.008) | (0.056) | (0.006) | (0.071) |
| Study month[a] | | | | | | |
| February | 0.113 | 0.051 | | | | |
| | (0.245) | (0.128) | | | | |
| March | -0.636*** | -0.387** | | | | |
| | (0.208) | (0.187) | | | | |
| April | -0.167 | 0.416 | | | | |
| | (0.233) | (0.259) | | | | |
| November | 0.019 | 0.37*** | | -2.306*** | 0.492 | |
| | (0.163) | (0.137) | | (0.125) | (0.339) | |
| December | 0.182 | -0.421** | 0.29*** | -2.179*** | -0.026 | -0.064 |
| | (0.125) | (0.16) | (0.11) | (0.272) | (0.277) | (0.211) |
| Countries [*Ref: Mozambique*] | | | | | | |
| Malawi | -0.181 | | | | | |
| | (0.207) | | | | | |
| Rwanda | -0.236 | | | | | |
| | (0.169) | | | | | |
| Uganda | -0.783** | | | | | |
| | (0.364) | | | | | |
| Zambia | 0.06 | | | | | |
| | (0.164) | | | | | |
| Observations | 19709 | 2594 | 4031 | 4777 | 4068 | 4239 |
| R-squared | 0.05 | 0.124 | 0.057 | 0.037 | 0.084 | 0.06 |

Note: Standard errors in parentheses

*** $p < 0.01$

** $p < 0.05$

* $p < 0.1$

[a]Ref categories; January (Pooled, Mozambique, Rwanda, Malawi, Uganda), November (Zambia).

**Table 8. OLS regression results for WDDS–resources domain (sole/joint decision in $\geq 1$ asset sale, purchase, and transfer).**

| | (1) | (2) | (3) | (4) | (5) | (6) |
|---|---|---|---|---|---|---|
| | All | Mozambique | Rwanda | Malawi | Uganda | Zambia |
| Input in asset transctns | 0.048 | 0.319** | 0.449*** | -0.104 | -0.152 | -0.122 |
| | (0.075) | (0.125) | (0.114) | (0.112) | (0.132) | (0.095) |
| SES index | -0.018 | 0.006 | 0.734 | -0.287** | -0.677 | -10.677** |
| | (0.108) | (0.306) | (10.049) | (0.128) | (0.565) | (0.641) |
| SES index squared | 0.019 | 0.131 | 0.227 | 0.023 | 0.116 | -0.944** |
| | (0.015) | (0.209) | (0.343) | (0.015) | (0.076) | (0.391) |
| Men's age | 0.006*** | 0.007* | 0.002 | 0.006* | 0.009*** | 0.003 |
| | (0.002) | (0.004) | (0.002) | (0.003) | (0.003) | (0.003) |
| Women's age | -0.011*** | -0.013*** | -0.01*** | -0.016*** | -0.012*** | -0.003 |
| | (0.002) | (0.004) | (0.004) | (0.003) | (0.003) | (0.003) |
| Women's education | 0.04*** | 0.02 | 0.128*** | 0.088** | 0.031*** | 0.041*** |
| | (0.009) | (0.06) | (0.034) | (0.038) | (0.01) | (0.013) |
| Household size | 0.033** | 0.048* | 0.041 | 0.037* | 0.014 | 0.046*** |
| | (0.013) | (0.027) | (0.031) | (0.02) | (0.019) | (0.013) |
| Study location | -0.014*** | 0.061*** | 0.018** | 0.021 | -0.028*** | -0.073 |
| | (0.005) | (0.013) | (0.008) | (0.056) | (0.006) | (0.071) |
| Study month[a] | | | | | | |
| February | 0.088 | -0.023 | | | | |
| | (0.246) | (0.136) | | | | |
| March | -0.648*** | -0.433** | | | | |
| | (0.207) | (0.191) | | | | |
| April | -0.178 | 0.376 | | | | |
| | (0.232) | (0.258) | | | | |
| November | 0.008 | 0.32** | | -2.302*** | 0.424 | |
| | (0.162) | (0.135) | | (0.122) | (0.332) | |
| December | 0.179 | -0.444*** | 0.322*** | -2.162*** | -0.078 | -0.048 |
| | (0.124) | (0.161) | (0.109) | (0.266) | (0.285) | (0.213) |
| Countries [Ref: Mozambique] | | | | | | |
| Malawi | -0.158 | | | | | |
| | (0.208) | | | | | |
| Rwanda | -0.235 | | | | | |
| | (0.169) | | | | | |
| Uganda | -0.798** | | | | | |
| | (0.368) | | | | | |
| Zambia | 0.056 | | | | | |
| | (0.163) | | | | | |
| Observations | 19576 | 2559 | 4013 | 4746 | 4052 | 4206 |
| R-squared | 0.05 | 0.117 | 0.054 | 0.037 | 0.088 | 0.06 |

Note: Standard errors in parentheses

*** p<0.01

** p<0.05

* p<0.1

[a]Ref categories; January (Pooled, Mozambique, Rwanda, Malawi, Uganda), November (Zambia).

**Table 9. OLS regression results for WDDS–resources domain (input in $\geq$ 1 credit source).**

| | (1) | (2) | (3) | (4) | (5) | (6) |
|---|---|---|---|---|---|---|
| | **All** | **Mozambique** | **Rwanda** | **Malawi** | **Uganda** | **Zambia** |
| Input in credit source decs | -0.04 | 0.210 | 0.249*** | -0.09 | -0.2** | -0.056 |
| | (0.056) | (0.192) | (0.069) | (0.077) | (0.094) | (0.062) |
| SES index | -0.009 | 0.031 | 0.618 | -0.287** | -0.682 | -10.682** |
| | (0.108) | (0.317) | (10.028) | (0.129) | (0.556) | (0.646) |
| SES index squared | 0.017 | 0.146 | 0.195 | 0.022 | 0.117 | -0.956** |
| | (0.015) | (0.22) | (0.337) | (0.014) | (0.075) | (0.393) |
| Men's age | 0.006*** | 0.008** | 0.003 | 0.006* | 0.009*** | 0.003 |
| | (0.002) | (0.004) | (0.002) | (0.003) | (0.003) | (0.003) |
| Women's age | -0.011*** | -0.012*** | -0.01** | -0.016*** | -0.011*** | -0.003 |
| | (0.002) | (0.004) | (0.004) | (0.003) | (0.004) | (0.003) |
| Women's education | 0.041*** | 0.013 | 0.129*** | 0.088** | 0.034*** | 0.041*** |
| | (0.009) | (0.059) | (0.033) | (0.039) | (0.01) | (0.013) |
| Household size | 0.032** | 0.048* | 0.041 | 0.036* | 0.014 | 0.046*** |
| | (0.013) | (0.028) | (0.031) | (0.02) | (0.018) | (0.013) |
| Study location | -0.015*** | 0.062*** | 0.02** | 0.02 | -0.028*** | -0.077 |
| | (0.005) | (0.013) | (0.008) | (0.055) | (0.006) | (0.072) |
| Study month[a] | | | | | | |
| February | 0.088 | -0.023 | | | | |
| | (0.246) | (0.136) | | | | |
| March | -0.648*** | -0.433** | | | | |
| | (0.207) | (0.191) | | | | |
| April | -0.178 | 0.376 | | | | |
| | (0.232) | (0.258) | | | | |
| November | 0.008 | 0.32** | | -2.302*** | 0.424 | |
| | (0.162) | (0.135) | | (0.122) | (0.332) | |
| December | 0.179 | -0.444*** | 0.322*** | -2.162*** | -0.078 | -0.048 |
| | (0.124) | (0.161) | (0.109) | (0.266) | (0.285) | (0.213) |
| Countries [*Ref: Mozambique*] | | | | | | |
| Malawi | -0.158 | | | | | |
| | (0.208) | | | | | |
| Rwanda | -0.235 | | | | | |
| | (0.169) | | | | | |
| Uganda | -0.798** | | | | | |
| | (0.368) | | | | | |
| Zambia | 0.056 | | | | | |
| | (0.163) | | | | | |
| Observations | 19576 | 2559 | 4013 | 4746 | 4052 | 4206 |
| R-squared | 0.05 | 0.117 | 0.054 | 0.037 | 0.088 | 0.06 |

Note: Standard errors in parentheses

*** p<0.01

** p<0.05

* p<0.1

[a]Ref categories; January (Pooled, Mozambique, Rwanda, Malawi, Uganda), November (Zambia).

**Table 10. OLS regression results for WDDS–regression results for WDDS–income domain (Input in at least 1 income decision).**

| | (1) | (2) | (3) | (4) | (5) | (6) |
|---|---|---|---|---|---|---|
| | All | Mozambique | Rwanda | Malawi | Uganda | Zambia |
| Input in ≥1 income domain | -0.036 | -0.042 | 0.098 | 0.195 | -0.122 | -0.064 |
| | (0.114) | (0.141) | (0.233) | (0.142) | (0.184) | (0.117) |
| SES index | -0.021 | 0.017 | 0.681 | -0.288** | -0.701 | -10.673** |
| | (0.109) | (0.318) | (10.069) | (0.13) | (0.572) | (0.64) |
| SES index squared | 0.02 | 0.145 | 0.215 | 0.022 | 0.12 | -0.938** |
| | (0.015) | (0.219) | (0.349) | (0.015) | (0.077) | (0.391) |
| Men's age | 0.005*** | 0.007** | 0.003 | 0.006* | 0.008*** | 0.003 |
| | (0.002) | (0.004) | (0.002) | (0.003) | (0.003) | (0.003) |
| Women's age | -0.011*** | -0.012*** | -0.01*** | -0.016*** | -0.012*** | -0.003 |
| | (0.002) | (0.004) | (0.004) | (0.003) | (0.003) | (0.003) |
| Women's education | 0.041*** | 0.012 | 0.128*** | 0.088** | 0.031*** | 0.041*** |
| | (0.009) | (0.06) | (0.034) | (0.038) | (0.01) | (0.013) |
| Household size | 0.032** | 0.051* | 0.047 | 0.035* | 0.013 | 0.046*** |
| | (0.013) | (0.027) | (0.032) | (0.02) | (0.019) | (0.013) |
| Study location | -0.014*** | 0.061*** | 0.019** | 0.024 | -0.027*** | -0.076 |
| | (0.005) | (0.013) | (0.008) | (0.056) | (0.006) | (0.072) |
| Study month[a] | | | | | | |
| February | 0.109 | 0.012 | | | | |
| | (0.247) | (0.131) | | | | |
| March | -0.636*** | -0.407** | | | | |
| | (0.207) | (0.187) | | | | |
| April | -0.169 | 0.384 | | | | |
| | (0.231) | (0.255) | | | | |
| November | 0.018 | 0.329** | | -2.288*** | 0.417 | |
| | (0.164) | (0.137) | | (0.122) | (0.306) | |
| December | 0.18 | -0.441*** | 0.298** | -2.17*** | -0.159 | -0.06 |
| | (0.126) | (0.162) | (0.114) | (0.271) | (0.254) | (0.212) |
| Countries [*Ref: Mozambique*] | | | | | | |
| Malawi | -0.181 | | | | | |
| | (0.209) | | | | | |
| Rwanda | -0.239 | | | | | |
| | (0.168) | | | | | |
| Uganda | -0.783** | | | | | |
| | (0.366) | | | | | |
| Zambia | 0.057 | | | | | |
| | (0.162) | | | | | |
| Observations | 19541 | 2591 | 3973 | 4744 | 3996 | 4237 |
| R-squared | 0.05 | 0.12 | 0.046 | 0.036 | 0.085 | 0.059 |

Note: Standard errors in parentheses

*** p<0.01

** p<0.05

* p<0.1

[a]Ref categories; January (Pooled, Mozambique, Rwanda, Malawi, Uganda), November (Zambia).

**Table 11. OLS regression results for WDDS–leadership domain (group membership).**

| | (1) | (2) | (3) | (4) | (5) | (6) |
|---|---|---|---|---|---|---|
| | **All** | **Mozambique** | **Rwanda** | **Malawi** | **Uganda** | **Zambia** |
| Membership of $\geq 1$ group | 0.008 | 0.02 | 0.317** | 0.057 | -0.209* | 0.019 |
| | (0.064) | (0.094) | (0.124) | (0.077) | (0.116) | (0.086) |
| SES index | -0.019 | 0.017 | 0.557 | -0.282** | -0.549 | -10.688** |
| | (0.109) | (0.34) | (10.029) | (0.131) | (0.561) | (0.669) |
| SES index squared | 0.021 | 0.14 | 0.176 | 0.022 | 0.099 | -0.944** |
| | (0.015) | (0.227) | (0.338) | (0.015) | (0.076) | (0.404) |
| Men's age | 0.005*** | 0.007* | 0.002 | 0.006* | 0.008*** | 0.003 |
| | (0.002) | (0.004) | (0.002) | (0.003) | (0.003) | (0.003) |
| Women's age | -0.012*** | -0.011*** | -0.011*** | -0.016*** | -0.012*** | -0.004 |
| | (0.002) | (0.004) | (0.004) | (0.003) | (0.004) | (0.003) |
| Women's education | 0.04*** | 0.075 | 0.109*** | 0.088** | 0.031*** | 0.04*** |
| | (0.009) | (0.064) | (0.033) | (0.039) | (0.01) | (0.013) |
| Household size | 0.032** | 0.054** | 0.05 | 0.037* | 0.016 | 0.045*** |
| | (0.014) | (0.026) | (0.032) | (0.02) | (0.019) | (0.013) |
| Study location | -0.012** | 0.077*** | 0.021** | 0.021 | -0.026*** | -0.076 |
| | (0.005) | (0.016) | (0.008) | (0.056) | (0.007) | (0.073) |
| Study month[a] | | | | | | |
| February | 0.114 | 0.233 | | | | |
| | (0.263) | (0.149) | | | | |
| March | -0.605*** | -0.204 | | | | |
| | (0.201) | (0.197) | | | | |
| April | -0.211 | 0.603** | | | | |
| | (0.232) | (0.295) | | | | |
| November | 0.015 | 0.421*** | | -2.281*** | 0.476 | |
| | (0.163) | (0.152) | | (0.123) | (0.319) | |
| December | 0.178 | -0.349* | 0.289** | -2.156*** | -0.054 | -0.048 |
| | (0.124) | (0.177) | (0.115) | (0.269) | (0.268) | (0.215) |
| Countries [*Ref: Mozambique*] | | | | | | |
| Malawi | -0.186 | | | | | |
| | (0.207) | | | | | |
| Rwanda | -0.235 | | | | | |
| | (0.166) | | | | | |
| Uganda | -0.816** | | | | | |
| | (0.367) | | | | | |
| Zambia | 0.081 | | | | | |
| | (0.161) | | | | | |
| Observations | 18913 | 2201 | 3895 | 4730 | 3927 | 4160 |
| R-squared | 0.049 | 0.156 | 0.054 | 0.036 | 0.08 | 0.059 |

Note: Standard errors in parentheses

*** p<0.01

** p<0.05

* p<0.1

[a]Ref categories; January (Pooled, Mozambique, Rwanda, Malawi, Uganda), November (Zambia).

**Table 12. OLS regression results for WDDS–leadership domain (comfortable speaking in public in ≥1 context).**

| | (1) | (2) | (3) | (4) | (5) | (6) |
|---|---|---|---|---|---|---|
| | All | Mozambique | Rwanda | Malawi | Uganda | Zambia |
| Public speaking | 0.247*** | 0.223** | 0.469*** | -0.082 | -0.055 | 0.096 |
| | (0.055) | (0.112) | (0.101) | (0.076) | (0.125) | (0.086) |
| SES index | -0.013 | 0.021 | 0.82 | -0.279** | -0.661 | -10.671** |
| | (0.108) | (0.312) | (1) | (0.127) | (0.569) | (0.643) |
| SES index squared | 0.018 | 0.146 | 0.251 | 0.022 | 0.114 | -0.939** |
| | (0.015) | (0.214) | (0.329) | (0.014) | (0.076) | (0.393) |
| Men's age | 0.006*** | 0.007* | 0.003 | 0.006* | 0.009*** | 0.003 |
| | (0.002) | (0.004) | (0.002) | (0.003) | (0.003) | (0.003) |
| Women's age | -0.011*** | -0.013*** | -0.01*** | -0.016*** | -0.012*** | -0.004 |
| | (0.002) | (0.004) | (0.004) | (0.003) | (0.003) | (0.003) |
| Women's education | 0.04*** | 0.012 | 0.125*** | 0.089** | 0.031*** | 0.04*** |
| | (0.009) | (0.06) | (0.033) | (0.039) | (0.01) | (0.013) |
| Household size | 0.032** | 0.05* | 0.052* | 0.038* | 0.014 | 0.045*** |
| | (0.013) | (0.027) | (0.03) | (0.02) | (0.019) | (0.013) |
| Study location | -0.014*** | 0.059*** | 0.019** | 0.02 | -0.027*** | -0.072 |
| | (0.005) | (0.013) | (0.008) | (0.056) | (0.006) | (0.072) |
| Study month[a] | | | | | | |
| February | 0.122 | -0.056 | | | | |
| | (0.243) | (0.145) | | | | |
| March | -0.616*** | -0.47** | | | | |
| | (0.205) | (0.21) | | | | |
| April | -0.158 | 0.296 | | | | |
| | (0.23) | (0.273) | | | | |
| November | 0.028 | 0.244 | | -2.283*** | 0.346 | |
| | (0.162) | (0.161) | | (0.123) | (0.313) | |
| December | 0.191 | -0.509*** | 0.329*** | -2.145*** | -0.153 | -0.052 |
| | (0.124) | (0.169) | (0.109) | (0.271) | (0.251) | (0.213) |
| Countries [Ref: Mozambique] | | | | | | |
| Malawi | -0.181 | | | | | |
| | (0.205) | | | | | |
| Rwanda | -0.231 | | | | | |
| | (0.166) | | | | | |
| Uganda | -0.81** | | | | | |
| | (0.363) | | | | | |
| Zambia | 0.057 | | | | | |
| | (0.161) | | | | | |
| Observations | 19670 | 2594 | 4020 | 4761 | 4067 | 4228 |
| R-squared | 0.051 | 0.123 | 0.06 | 0.036 | 0.083 | 0.06 |

Note: Standard errors in parentheses

*** p<0.01

** p<0.05

* p<0.1

[a]Ref categories; January (Pooled, Mozambique, Rwanda, Malawi, Uganda), November (Zambia).

**Table 13. OLS regression for WDDS–time domain (Non-excessive workload [<10.5hrs in 24hrs]).**

| | (1) | (2) | (3) | (4) | (5) | (6) |
|---|---|---|---|---|---|---|
| | **All** | **Mozambique** | **Rwanda** | **Malawi** | **Uganda** | **Zambia** |
| Non-excessive workload | -0.098 | -0.327* | -0.093 | -0.017 | -0.089 | -0.052 |
| | (0.052) | (0.143) | (0.079) | (0.051) | (0.088) | (0.082) |
| SES index | -0.046 | -0.004 | 0.592 | -0.268** | -0.935** | -10.748** |
| | (0.108) | (0.326) | (10.14) | (0.123) | (0.439) | (0.68) |
| SES index squared | 0.02 | 0.122 | 0.191 | 0.021 | 0.148** | -0.987** |
| | (0.015) | (0.213) | (0.371) | (0.014) | (0.06) | (0.411) |
| Men's age | 0.005*** | 0.007* | 0.002 | 0.006* | 0.008*** | 0.003 |
| | (0.002) | (0.004) | (0.002) | (0.003) | (0.003) | (0.003) |
| Women's age | -0.012*** | -0.009** | -0.01*** | -0.016*** | -0.013*** | -0.003 |
| | (0.002) | (0.004) | (0.004) | (0.003) | (0.004) | (0.003) |
| Women's education | 0.039*** | 0.06 | 0.119*** | 0.085** | 0.029*** | 0.04*** |
| | (0.009) | (0.064) | (0.036) | (0.039) | (0.01) | (0.014) |
| Household size | 0.042*** | 0.044 | 0.052 | 0.045** | 0.026 | 0.046*** |
| | (0.012) | (0.027) | (0.034) | (0.023) | (0.017) | (0.013) |
| Study location | -0.013** | 0.074*** | 0.02** | 0.026 | -0.027*** | -0.079 |
| | (0.005) | (0.016) | (0.008) | (0.057) | (0.007) | (0.072) |
| Study month[a] | | | | | | |
| February | 0.047 | 0.105 | | | | |
| | (0.263) | (0.158) | | | | |
| March | -0.616*** | -0.314 | | | | |
| | (0.203) | (0.202) | | | | |
| April | -0.281 | 0.415 | | | | |
| | (0.236) | (0.304) | | | | |
| November | -0.014 | 0.293* | | -2.31*** | 0.342 | |
| | (0.167) | (0.162) | | (0.127) | (0.286) | |
| December | 0.157 | -0.431*** | 0.278** | -2.202*** | -0.163 | -0.048 |
| | (0.13) | (0.161) | (0.118) | (0.274) | (0.182) | (0.21) |
| Countries [*Ref: Mozambique*] | | | | | | |
| Malawi | -0.234 | | | | | |
| | (0.209) | | | | | |
| Rwanda | -0.275 | | | | | |
| | (0.17) | | | | | |
| Uganda | -0.731** | | | | | |
| | (0.359) | | | | | |
| Zambia | -0.004 | | | | | |
| | (0.162) | | | | | |
| Observations | 18117 | 2100 | 3681 | 4569 | 3754 | 4013 |
| R-squared | 0.054 | 0.156 | 0.047 | 0.037 | 0.091 | 0.061 |

Note: Standard errors in parentheses

*** p<0.01

** p<0.05

* p<0.1

[a]Ref categories; January (Pooled, Mozambique, Rwanda, Malawi, Uganda), November (Zambia).

**Table 14. OLS regression for WDDS–time domain (satisfaction with leisure time).**

| | (1) | (2) | (3) | (4) | (5) | (6) |
|---|---|---|---|---|---|---|
| | **All** | **Mozambique** | **Rwanda** | **Malawi** | **Uganda** | **Zambia** |
| Leisure time | -0.006 | -0.064 | -0.011 | 0.146* | -0.034 | -0.011 |
| | (0.075) | (0.145) | (0.103) | (0.083) | (0.104) | (0.092) |
| SES index | -0.018 | -0.013 | 0.65 | -0.288** | -0.684 | -10.66** |
| | (0.109) | (0.319) | (10.086) | (0.128) | (0.564) | (0.644) |
| SES index squared | 0.019 | 0.127 | 0.197 | 0.022 | 0.117 | -0.93** |
| | (0.015) | (0.219) | (0.355) | (0.014) | (0.075) | (0.392) |
| Men's age | 0.006*** | 0.007** | 0.003 | 0.006* | 0.008*** | 0.003 |
| | (0.002) | (0.004) | (0.002) | (0.003) | (0.003) | (0.003) |
| Women's age | -0.011*** | -0.012*** | -0.01** | -0.016*** | -0.012*** | -0.003 |
| | (0.002) | (0.004) | (0.004) | (0.003) | (0.003) | (0.003) |
| Women's education | 0.04*** | 0.013 | 0.126*** | 0.088** | 0.031*** | 0.04*** |
| | (0.009) | (0.06) | (0.034) | (0.039) | (0.01) | (0.013) |
| Household size | 0.033** | 0.051* | 0.047 | 0.037* | 0.014 | 0.046*** |
| | (0.013) | (0.027) | (0.032) | (0.02) | (0.019) | (0.013) |
| Study location | -0.014*** | 0.06*** | 0.018** | 0.02 | -0.027*** | -0.075 |
| | (0.005) | (0.013) | (0.008) | (0.056) | (0.006) | (0.07) |
| Study month[a] | | | | | | |
| February | 0.102 | 0.015 | | | | |
| | (0.244) | (0.131) | | | | |
| March | -0.644*** | -0.408** | | | | |
| | (0.207) | (0.188) | | | | |
| April | -0.172 | 0.383 | | | | |
| | (0.232) | (0.255) | | | | |
| November | 0.016 | 0.333** | | -2.266*** | 0.36 | |
| | (0.164) | (0.136) | | (0.126) | (0.32) | |
| December | 0.178 | -0.438*** | 0.293** | -2.144*** | -0.139 | -0.059 |
| | (0.125) | (0.164) | (0.113) | (0.272) | (0.257) | (0.208) |
| Countries [*Ref: Mozambique*] | | | | | | |
| Malawi | -0.185 | | | | | |
| | (0.209) | | | | | |
| Rwanda | -0.241 | | | | | |
| | (0.169) | | | | | |
| Uganda | -0.789** | | | | | |
| | (0.367) | | | | | |
| Zambia | 0.056 | | | | | |
| | (0.164) | | | | | |
| Observations | 19652 | 2587 | 4004 | 4763 | 4063 | 4235 |
| R-squared | 0.05 | 0.119 | 0.045 | 0.037 | 0.082 | 0.059 |

Note: Standard errors in parentheses

*** p<0.01

** p<0.05

* p<0.1

[a]Ref categories; January (Pooled, Mozambique, Rwanda, Malawi, Uganda), November (Zambia).

Empowerment in autonomy in at least one production activity was associated with a 18%-point increase in WDDS in the pooled analysis and Uganda appears to account for this significant association. Empowerment in input in at least two production decisions was associated with a 21%-point increase in WDDS in the pooled analysis and Rwanda appears to account for this association. Empowerment in public speaking was associated with a 2%-point increase in WDDS in the pooled analysis and Mozambique and Rwanda appear to account for these significant findings.

## Women's empowerment and food consumption

Focusing on the three empowerment indicators with significant associations with WDDS, the present study further examined which food groups within the WDDS that might account for these significant associations. Findings are presented in Tables 15–21 (see S12–S17 Tables for results from confirmatory regression analyses using marginal effects of the logistic regression). Findings show variations in food items that accounted for the relationship between the four empowerment domains and WDDS and these variations differ across the five SSA countries examined. These findings were also confirmed in the logistic regression models with marginal effects where coefficients were similar to those from the LPM regression analyses.

Results suggest that in the pooled analysis, empowerment in autonomy in at least one production activity was significantly and positively associated with a 2%-point increase in the likelihood of consumption of dairy and dairy products, eggs, and flesh protein, a 5%-point increase in the likelihood of consumption of vitamin A-rich leafy greens, and an 8%-point increase in the likelihood of consuming other vitamin A-rich fruits and vegetables. Empowerment through input in at least two domains of production decisions was significantly associated with about 2%-points increase in the likelihood of the consumption of grains and tubers, about a 3%-point increase in the likelihood of consuming dairy and dairy products, a 5%-point increase in the consumption of other fruits and vegetables as well as vitamin A-rich products. This implies that women who had input in at least two domains of food production decisions were more likely to consume these four out of the nine WDDS food groups in the pooled analysis. Further disaggregation by country suggests that in Uganda, empowerment in autonomy in production activities was associated with the increased likelihood of consuming legumes, vitamin A-rich dark-green leafy vegetables, and other fruits and vegetables including vitamin A-rich produce. In Rwanda, empowerment in input in at least two production decisions was associated increased likelihood of consuming grains and tubers, dairy and dairy products, flesh proteins, vitamin A-rich dark-green leafy vegetables, and other fruits and vegetables including vitamin A-rich products.

In the pooled analysis, findings suggest that other fruits and vegetables (2%-point increase in likelihood) including vitamin A-rich produce (6%-point increase in likelihood) might account for the association between empowerment in public speaking and improved WDDS. Further disaggregation suggests that Mozambique and Rwanda might account for these associations in the pooled analyses. In Mozambique, empowerment in public speaking was significantly associated with the likelihood of women consuming vitamin A-rich dark-green leafy vegetables, and other fruits and vegetables including vitamin A-rich produce. In Rwanda, empowerment in public speaking was significantly associated with the likelihood of women consuming grains and tubers, vitamin A-rich dark-green leafy vegetables, and other fruits and vegetables including vitamin A-rich produce.

## Discussion

The present study has found varying degrees of associations between four empowerment indicators and women's dietary diversity and food consumption, and these associations also varied

**Table 15.  LPM regression for food groups consumed–production domain (autonomy in $\geq$ 1 activity linked to production)–pooled.**

| | (1) | (2) | (3) | (4) | (5) | (6) | (7) | (8) | (9) |
|---|---|---|---|---|---|---|---|---|---|
| | Grain/ Roots | Legumes | Dairy | Organ meat | Eggs | Flesh protein | Vit A-rich dark-green leafy veg | Other vit A-rich frts/ vegs | Other frts/ vegs |
| Aut in prod decs | 0.013* | 0.025 | 0.028** | 0.003 | 0.024*** | 0.023** | 0.056*** | 0.088*** | 0.018 |
| | (0.007) | (0.011) | (0.009) | (0.004) | (0.008) | (0.012) | (0.013) | (0.013) | (0.013) |
| SES index | -0.003 | -0.029*** | 0.061*** | -0.013*** | 0.004 | 0.002 | 0.017 | -0.024** | -0.027** |
| | (0.006) | (0.01) | (0.009) | (0.004) | (0.005) | (0.011) | (0.012) | (0.012) | (0.012) |
| SES index squared | 0.002* | 0.002 | -0.006*** | 0.002*** | 0.001 | 0.006*** | -0.005*** | 0.004** | 0.013*** |
| | (0.001) | (0.002) | (0.001) | (0.001) | (0.001) | (0.002) | (0.002) | (0.002) | (0.002) |
| Men's age | 0.000 | 0.001*** | 0.001*** | 0.000 | 0.000** | 0.001** | 0.000 | 0.001 | 0.001*** |
| | (0.000) | (0.000) | (0.000) | (0.000) | (0.000) | (0.000) | (0.000) | (0.000) | (0.000) |
| Women's age | 0.000* | 0.000 | -0.001** | 0.000** | -0.002*** | -0.001*** | 0.000 | -0.005*** | -0.002*** |
| | (0.000) | (0.000) | (0.000) | (0.000) | (0.000) | (0.000) | (0.000) | (0.000) | (0.000) |
| Women's education | 0.001 | 0.003 | 0.011*** | 0.001** | 0.004*** | 0.007*** | -0.002 | 0.006*** | 0.011*** |
| | (0.001) | (0.002) | (0.001) | (0.000) | (0.001) | (0.002) | (0.002) | (0.002) | (0.002) |
| Household size | 0.002*** | -0.002 | 0.004*** | -0.001*** | 0.001 | 0.003** | 0.015*** | 0.007*** | 0.003** |
| | (0.001) | (0.001) | (0.001) | (0.000) | (0.001) | (0.002) | (0.002) | (0.002) | (0.002) |
| Study location | 0.000 | -0.001** | -0.003*** | 0.000*** | -0.002*** | -0.003*** | -0.004*** | 0.001** | -0.001** |
| | (0.000) | (0.000) | (0.000) | (0.000) | (0.000) | (0.001) | (0.001) | (0.001) | (0.001) |
| Study month | | | | | | | | | |
| February | -0.072*** | -0.112*** | -0.004 | 0.014** | 0.079*** | 0.172*** | 0.058* | 0.126*** | -0.208*** |
| | (0.012) | (0.034) | (0.011) | (0.006) | (0.018) | (0.031) | (0.032) | (0.036) | (0.031) |
| March | -0.084*** | -0.193*** | -0.017 | 0.022** | -0.025 | 0.167*** | -0.094** | -0.07* | -0.395*** |
| | (0.018) | (0.039) | (0.013) | (0.011) | (0.018) | (0.039) | (0.039) | (0.04) | (0.037) |
| April | -0.045*** | -0.122*** | -0.011 | 0.007 | 0.036 | 0.133*** | -0.056 | 0.044 | -0.175*** |
| | (0.016) | (0.044) | (0.017) | (0.007) | (0.028) | (0.043) | (0.043) | (0.045) | (0.043) |
| November | -0.001 | -0.027 | 0.034*** | 0.005 | -0.013 | -0.221*** | 0.12*** | 0.028 | 0.066*** |
| | (0.009) | (0.017) | (0.012) | (0.007) | (0.008) | (0.018) | (0.02) | (0.02) | (0.018) |
| December | 0.007 | -0.02* | 0.013 | 0.002 | 0.01** | -0.048*** | -0.022 | 0.154*** | 0.078*** |
| | (0.008) | (0.011) | (0.011) | (0.005) | (0.005) | (0.011) | (0.016) | (0.016) | (0.014) |
| Countries [*Ref: Mozambique*] | | | | | | | | | |
| Malawi | -0.085*** | 0.42*** | 0.187*** | 0.01 | -0.05*** | -0.217*** | -0.106*** | 0.234*** | -0.607*** |
| | (0.01) | (0.033) | (0.014) | (0.006) | (0.016) | (0.03) | (0.032) | (0.035) | (0.029) |
| Rwanda | -0.027*** | -0.064** | -0.016** | 0.014*** | -0.008 | 0.202*** | -0.101*** | 0.334*** | -0.619*** |
| | (0.004) | (0.031) | (0.007) | (0.004) | (0.015) | (0.028) | (0.027) | (0.031) | (0.025) |
| Uganda | -0.08*** | 0.485*** | -0.123*** | 0.035*** | -0.071*** | -0.137*** | -0.266*** | -0.032 | -0.658*** |
| | (0.021) | (0.044) | (0.028) | (0.012) | (0.022) | (0.046) | (0.049) | (0.05) | (0.045) |
| Zambia | -0.019*** | 0.033 | -0.013 | 0.035*** | 0.024 | 0.239*** | -0.244*** | 0.511*** | -0.538*** |
| | (0.005) | (0.032) | (0.01) | (0.005) | (0.017) | (0.029) | (0.03) | (0.032) | (0.027) |
| Observations | 19756 | 19756 | 19756 | 19756 | 19756 | 19756 | 19756 | 19756 | 19756 |
| R-squared | 0.011 | 0.198 | 0.041 | 0.009 | 0.038 | 0.096 | 0.068 | 0.111 | 0.054 |

Note: Standard errors in parentheses

*** p<0.01

** p<0.05

* p<0.1.

**Table 16. LPM regression for food groups consumed–production domain (input in $\geq 2$ productive decisions)–pooled.**

| | (1) | (2) | (3) | (4) | (5) | (6) | (7) | (8) | (9) |
|---|---|---|---|---|---|---|---|---|---|
| | Grain/ Roots | Legumes | Dairy | Organ meat | Eggs | Flesh protein | Vit A-rich dark-green leafy vegs | Other vit A-rich frts/ vegs | Other frts/ vegs |
| Input in prod decs | 0.017*** | 0.029 | 0.036** | 0.002 | 0.017 | 0.036 | 0.021** | 0.051*** | 0.056*** |
| | (0.005) | (0.009) | (0.007) | (0.002) | (0.004) | (0.009) | (0.01) | (0.01) | (0.009) |
| SES index | -0.002 | -0.027*** | 0.062*** | -0.012*** | 0.004 | 0.006 | 0.01 | -0.028** | -0.026** |
| | (0.006) | (0.01) | (0.009) | (0.004) | (0.005) | (0.011) | (0.013) | (0.012) | (0.012) |
| SES index squared | 0.002** | 0.002 | -0.007*** | 0.002*** | 0.002** | 0.007*** | -0.005** | 0.005** | 0.013*** |
| | (0.001) | (0.002) | (0.001) | (0.001) | (0.001) | (0.002) | (0.002) | (0.002) | (0.002) |
| Men's age | 0.000 | 0.001*** | 0.001*** | 0.000 | 0.000** | 0.001** | 0.000 | 0.000 | 0.001*** |
| | (0.000) | (0.000) | (0.000) | (0.000) | (0.000) | (0) | (0.000) | (0.000) | (0.000) |
| Women's age | 0.000* | 0.000 | -0.001*** | 0.000** | -0.002*** | -0.001*** | 0.000 | -0.005*** | -0.003*** |
| | (0.000) | (0.000) | (0.000) | (0.000) | (0.000) | (0) | (0.000) | (0.000) | (0.000) |
| Women's education | 0.001 | 0.002 | 0.01*** | 0.001 | 0.003*** | 0.006*** | -0.001 | 0.004** | 0.011*** |
| | (0.001) | (0.002) | (0.001) | (0.000) | (0.001) | (0.002) | (0.002) | (0.002) | (0.002) |
| Household size | 0.002** | -0.001 | 0.003** | -0.001** | 0.002* | 0.003** | 0.014*** | 0.006*** | 0.003* |
| | (0.001) | (0.002) | (0.001) | (0.000) | (0.001) | (0.002) | (0.002) | (0.002) | (0.002) |
| Study location | -0.001** | -0.001 | -0.003*** | 0.000** | -0.003*** | -0.004*** | -0.004*** | 0.002** | -0.001** |
| | (0.000) | (0.000) | (0.000) | (0.000) | (0.000) | (0.001) | (0.001) | (0.001) | (0.001) |
| Study month | | | | | | | | | |
| February | -0.071*** | -0.121*** | -0.002 | 0.012** | 0.078*** | 0.179*** | 0.07** | 0.143*** | -0.212*** |
| | (0.012) | (0.034) | (0.011) | (0.006) | (0.018) | (0.032) | (0.033) | (0.036) | (0.031) |
| March | -0.082*** | -0.199*** | -0.013 | 0.021* | -0.027 | 0.184*** | -0.092** | -0.05 | -0.389*** |
| | (0.018) | (0.039) | (0.013) | (0.011) | (0.018) | (0.039) | (0.04) | (0.04) | (0.037) |
| April | -0.044*** | -0.129*** | -0.011 | 0.006 | 0.023 | 0.139*** | -0.062 | 0.052 | -0.178*** |
| | (0.016) | (0.044) | (0.017) | (0.007) | (0.028) | (0.044) | (0.044) | (0.046) | (0.043) |
| November | 0 | -0.04** | 0.028** | 0.002 | -0.022*** | -0.22*** | 0.117*** | 0.021 | 0.058*** |
| | (0.009) | (0.018) | (0.012) | (0.007) | (0.008) | (0.018) | (0.021) | (0.02) | (0.019) |
| December | 0.009 | -0.031*** | 0.008 | 0.000 | 0.004 | -0.05*** | -0.029* | 0.141*** | 0.071*** |
| | (0.009) | (0.011) | (0.011) | (0.005) | (0.005) | (0.011) | (0.016) | (0.016) | (0.014) |
| Countries [*Ref*: *Mozambique*] | | | | | | | | | |
| Malawi | -0.088*** | 0.418*** | 0.185*** | 0.011 | -0.054*** | -0.213*** | -0.102*** | 0.228*** | -0.616*** |
| | (0.01) | (0.033) | (0.014) | (0.007) | (0.016) | (0.03) | (0.033) | (0.035) | (0.03) |
| Rwanda | -0.036*** | -0.066** | -0.025*** | 0.015*** | -0.017 | 0.198*** | -0.084*** | 0.339*** | -0.63*** |
| | (0.004) | (0.031) | (0.007) | (0.004) | (0.015) | (0.028) | (0.028) | (0.031) | (0.025) |
| Uganda | -0.089*** | 0.475*** | -0.125*** | 0.031** | -0.076*** | -0.153*** | -0.236*** | -0.026 | -0.668*** |
| | (0.022) | (0.045) | (0.028) | (0.012) | (0.022) | (0.047) | (0.05) | (0.051) | (0.046) |
| Zambia | -0.026*** | 0.027 | -0.023** | 0.036*** | 0.017 | 0.236*** | -0.231*** | 0.511*** | -0.553*** |
| | (0.006) | (0.033) | (0.01) | (0.005) | (0.017) | (0.03) | (0.031) | (0.032) | (0.027) |
| Observations | 19303 | 19303 | 19303 | 19303 | 19303 | 19303 | 19303 | 19303 | 19303 |
| R-squared | 0.012 | 0.202 | 0.045 | 0.009 | 0.044 | 0.1 | 0.067 | 0.113 | 0.055 |

Note: Standard errors in parentheses

*** p<0.01

** p<0.05

* p<0.1.

**Table 17. LPM regression for food groups consumed–production domain (input in $\geq 2$ productive decisions)–Uganda.**

| | (1) | (2) | (3) | (4) | (5) | (6) | (7) | (8) | (9) |
|---|---|---|---|---|---|---|---|---|---|
| | Grain/ Roots | Legumes | Dairy | Organ meat | Eggs | Flesh protein | Vit A-rich dark-green leafy vegs | Other vit A-rich frts/ vegs | Other frts/ vegs |
| Aut in production decs | -0.001 | 0.069 | 0.028** | 0.01 | 0.055 | 0.026** | 0.102*** | 0.118*** | 0.076*** |
| | (0.012) | (0.022) | (0.019) | (0.007) | (0.017) | (0.024) | (0.024) | (0.026) | (0.024) |
| SES index | 0.05* | -0.135*** | -0.037 | -0.041** | 0.01 | -0.145* | 0.183*** | -0.162** | -0.238*** |
| | (0.03) | (0.044) | (0.048) | (0.018) | (0.03) | (0.077) | (0.07) | (0.074) | (0.077) |
| SES index squared | -0.006 | 0.018*** | 0.008 | 0.006** | 0.001 | 0.028** | -0.028*** | 0.023** | 0.045*** |
| | (0.004) | (0.006) | (0.007) | (0.003) | (0.004) | (0.011) | (0.01) | (0.01) | (0.011) |
| Men's age | 0.001* | 0.002** | 0.002*** | 0.000 | 0.001** | 0.002** | -0.001 | 0.000 | 0.002*** |
| | (0) | (0.001) | (0.001) | (0.000) | (0.000) | (0.001) | (0.001) | (0.001) | (0.001) |
| Women's age | -0.001** | 0.001 | -0.001 | 0.000 | -0.002*** | -0.002*** | 0.001 | -0.005*** | -0.003*** |
| | (0.000) | (0.001) | (0.001) | (0.000) | (0.000) | (0.001) | (0.001) | (0.001) | (0.001) |
| Women's education | 0.000 | 0.002 | 0.009*** | 0.001 | 0.004*** | 0.006*** | -0.004* | 0.003 | 0.01*** |
| | (0.001) | (0.002) | (0.001) | (0.000) | (0.001) | (0.002) | (0.002) | (0.002) | (0.002) |
| Household size | 0.002** | -0.009*** | 0.000 | -0.002*** | -0.001 | 0.002 | 0.019*** | 0.006** | -0.001 |
| | (0.001) | (0.002) | (0.002) | (0.001) | (0.001) | (0.002) | (0.003) | (0.003) | (0.002) |
| Study location | -0.001*** | -0.002*** | -0.003*** | -0.001*** | -0.003*** | -0.006*** | -0.007*** | -0.001 | -0.004*** |
| | (0.000) | (0.001) | (0.001) | (0.000) | (0.000) | (0.001) | (0.001) | (0.001) | (0.001) |
| Study month [*Ref: November*] | | | | | | | | | |
| December | -0.073*** | -0.233*** | 0.118*** | 0.000 | 0.010 | 0.208*** | 0.289*** | -0.629*** | 0.128 |
| | (0.009) | (0.017) | (0.014) | (0.003) | (0.010) | (0.019) | (0.019) | (0.02) | (0.093) |
| Observations | 4079 | 4079 | 4079 | 4079 | 4079 | 4079 | 4079 | 4079 | 4079 |
| R-squared | 0.008 | 0.015 | 0.024 | 0.013 | 0.043 | 0.04 | 0.047 | 0.028 | 0.051 |

Note: Standard errors are in parentheses

*** p<0.01

** p<0.05

* p<0.1.

across the five SSA countries examined. Findings suggest that while examining the relationship between empowerment and increases in dietary diversity is of importance, there is need to go further and examine within the dietary diversity measures, which food groups are responsible for the changes in the scores. This is important because evidence suggests that just as women's empowerment is heavily influenced by prevailing social and economic contextual factors [50, 66, 67], consumption of different food groups is also influenced by these factors [68]. For instance, a study in rural Bangladesh found that husband's occupation, women's education levels, and higher socioeconomic status was linked to consumption of nutrient-rich food items for pregnant women [69]. A qualitative study by Onah (2020) [38] found that households use a mix of food production and purchase for consumption where more nutrient-rich food items are largely purchased, and women's limited income restricts them from full empowerment in ensuring improved dietary diversity. Hence, empowerment may give women greater autonomy to improve their dietary diversity through the purchase of some of the more expensive items (flesh proteins, fish, dairy, other fruits and vegetables) which are not typically produced on one's own farm but are rich sources of micronutrients. This might also explain why different empowerment indicators affect some foods groups but not others. The role of other empowerment indicators towards the consumption of specific food items might shed more

**Table 18. LPM regression for food groups consumed–production domain (Input in ≥ 2 productive decisions)—Rwanda.**

| | (1) | (2) | (3) | (4) | (5) | (6) | (7) | (8) | (9) |
|---|---|---|---|---|---|---|---|---|---|
| | Grain/ Roots | Legumes | Dairy | Organ meat | Eggs | Flesh protein | Vit A-rich dark-green leafy vegs | Other vit A-rich frts/ vegs | Other frts/ vegs |
| Input in productive decs | 0.044*** | 0.047 | 0.072*** | 0.005 | 0.003 | 0.079*** | 0.054*** | 0.116*** | 0.106*** |
| | (0.01) | (0.013) | (0.011) | (0.006) | (0.005) | (0.011) | (0.017) | (0.017) | (0.014) |
| SES index | 0.071 | -0.29** | 0.275** | 0.166** | 0.01 | 0.327** | 0.014 | -0.31* | 0.422*** |
| | (0.089) | (0.14) | (0.129) | (0.066) | (0.037) | (0.141) | (0.163) | (0.171) | (0.144) |
| SES index squared | 0.027 | -0.074 | 0.039 | 0.06*** | 0.001 | 0.091** | 0.003 | -0.088 | 0.151*** |
| | (0.03) | (0.047) | (0.043) | (0.022) | (0.013) | (0.047) | (0.056) | (0.058) | (0.049) |
| Men's age | 0.000 | 0.000 | 0.000 | 0.000 | 0.000 | 0.000 | 0.001* | 0.001 | 0.000 |
| | (0.000) | (0.000) | (0.000) | (0.000) | (0.000) | (0.000) | (0.001) | (0.001) | (0.001) |
| Women's age | 0.000 | 0.000 | -0.001 | 0.001** | -0.001*** | 0.001 | -0.003*** | -0.005*** | -0.002*** |
| | (0.000) | (0.001) | (0.001) | (0.000) | (0.000) | (0.001) | (0.001) | (0.001) | (0.001) |
| Women's education | 0.01** | 0.003 | 0.021*** | 0.003 | 0.001 | 0.000 | 0.029*** | 0.023*** | 0.024*** |
| | (0.004) | (0.005) | (0.005) | (0.002) | (0.002) | (0.005) | (0.007) | (0.007) | (0.006) |
| Household size | -0.001 | 0.004 | 0.015*** | -0.001 | 0.002 | 0.002 | 0.001 | 0.004 | 0.005 |
| | (0.002) | (0.003) | (0.003) | (0.001) | (0.001) | (0.003) | (0.004) | (0.004) | (0.003) |
| Study location | 0.001** | 0.002*** | -0.003*** | 0.000 | 0.000 | 0.003*** | 0.007*** | 0.005*** | 0.006*** |
| | (0.000) | (0.001) | (0.001) | (0.000) | (0.000) | (0.001) | (0.001) | (0.001) | (0.001) |
| Study month [*Ref*: January] | | | | | | | | | |
| December | 0.013 | -0.015 | 0.015 | 0.001 | 0.016*** | -0.037*** | -0.017 | 0.165*** | 0.087*** |
| | (0.009) | (0.012) | (0.011) | (0.005) | (0.005) | (0.011) | (0.016) | (0.016) | (0.014) |
| Observations | 3848 | 3848 | 3848 | 3848 | 3848 | 3848 | 3848 | 3848 | 3848 |
| R-squared | 0.011 | 0.015 | 0.056 | 0.006 | 0.012 | 0.025 | 0.025 | 0.06 | 0.045 |

Note: Standard errors are in parentheses

*** p<0.01

** p<0.05

* p<0.1.

light on strategies that aim to use women's empowerment in improving nutrition outcomes including dietary diversity. For instance, empowerment in agricultural practices might be important for the consumption of green vegetables which women tend to cultivate in their own plots [8, 59, 70].

The findings that improved autonomy in production, and input in production decisions were significantly associated with improved dietary diversity for women is consistent with literature. Malapit et al. [50] in Nepal, Yimer Feiruz and Fanaye Tadesse [45] in Ethiopia, Ross et al. [73] in Ghana, and Sinharoy et al. [2] in Bangladesh all using the WEAI measure found that improvement in women's autonomy, and input in production was associated with improved women's dietary diversity. However, the present study goes a step further and has found that dairy products and vitamin A-rich fruits and vegetables including dark-green vegetables, and in one country flesh protein, might account for this association. Previous studies have not taken the analysis down to the level of specific nutrient-rich foods. Since vegetables and to some extent fruits are largely cultivated by the households and, when purchased, are not as expensive as other purchased food items, women's empowerment in production could possibly improve their consumption [37]. Another possible link in existing literature is the role of women's empowerment in crop diversity. This is consistent with De Pinto et al. [48]

**Table 19. Marginal effects of Logistic regression for food groups consumed–leadership domain (Comfortable speaking in public in ≥ 1 context)–pooled.**

| | (1) | (2) | (3) | (4) | (5) | (6) | (7) | (8) | (9) |
|---|---|---|---|---|---|---|---|---|---|
| | Grain/ Roots | Legumes | Dairy | Organ meat | Eggs | Flesh protein | Vit A-rich dark-green leafy vegs | Other vit A-rich frts/ vegs | Other frts/ vegs |
| Public speaking | 0.022 | -0.017 | -0.004 | 0.004 | -0.001 | 0.052 | 0.014 | 0.027** | 0.068*** |
| | (0.007) | (0.011) | (0.009) | (0.003) | (0.006) | (0.011) | (0.013) | (0.013) | (0.012) |
| SES index | -0.001 | -0.029*** | 0.061*** | -0.012*** | 0.004 | 0.004 | 0.017 | -0.024** | -0.026** |
| | (0.006) | (0.01) | (0.009) | (0.004) | (0.005) | (0.011) | (0.012) | (0.012) | (0.012) |
| SES index squared | 0.001 | 0.002 | -0.006*** | 0.002*** | 0.001 | 0.006*** | -0.005*** | 0.004** | 0.013*** |
| | (0.001) | (0.002) | (0.001) | (0.001) | (0.001) | (0.002) | (0.002) | (0.002) | (0.002) |
| Men's age | 0.000 | 0.001*** | 0.001*** | 0.000 | 0.000** | 0.001** | 0.000 | 0.001 | 0.001*** |
| | (0.000) | (0.000) | (0.000) | (0) | (0.000) | (0) | (0.000) | (0.000) | (0.000) |
| Women's age | 0.000* | 0.000 | -0.001** | 0.000** | -0.002*** | -0.001*** | 0.000 | -0.005*** | -0.002*** |
| | (0.000) | (0.000) | (0.000) | (0.000) | (0.000) | (0) | (0.000) | (0.000) | (0.000) |
| Women's education | 0.001 | 0.003 | 0.011*** | 0.001** | 0.004*** | 0.007*** | -0.002 | 0.005*** | 0.011*** |
| | (0.001) | (0.002) | (0.001) | (0.000) | (0.001) | (0.002) | (0.002) | (0.002) | (0.002) |
| Household size | 0.002*** | -0.002 | 0.004*** | -0.001*** | 0.001 | 0.003** | 0.015*** | 0.007*** | 0.003** |
| | (0.001) | (0.001) | (0.001) | (0.000) | (0.001) | (0.002) | (0.002) | (0.002) | (0.002) |
| Study location | 0.000 | -0.001** | -0.003*** | 0*** | -0.002*** | -0.003*** | -0.004*** | 0.001** | -0.001** |
| | (0.000) | (0.000) | (0.000) | (0.000) | (0.000) | (0.001) | (0.001) | (0.001) | (0.001) |
| Study month | | | | | | | | | |
| February | -0.069*** | -0.116*** | -0.003 | 0.013** | 0.081*** | 0.179*** | 0.065** | 0.132*** | -0.199*** |
| | (0.012) | (0.034) | (0.011) | (0.006) | (0.018) | (0.031) | (0.032) | (0.036) | (0.031) |
| March | -0.08*** | -0.198*** | -0.015 | 0.021* | -0.023 | 0.178*** | -0.085** | -0.061 | -0.381*** |
| | (0.018) | (0.039) | (0.013) | (0.011) | (0.018) | (0.039) | (0.04) | (0.039) | (0.037) |
| April | -0.041** | -0.123*** | -0.009 | 0.006 | 0.035 | 0.138*** | -0.054 | 0.042 | -0.168*** |
| | (0.016) | (0.044) | (0.017) | (0.007) | (0.028) | (0.043) | (0.044) | (0.045) | (0.043) |
| November | 0.004 | -0.029* | 0.035*** | 0.004 | -0.013 | -0.217*** | 0.122*** | 0.025 | 0.073*** |
| | (0.009) | (0.017) | (0.012) | (0.007) | (0.008) | (0.018) | (0.02) | (0.02) | (0.018) |
| December | 0.012 | -0.02* | 0.014 | 0.001 | 0.009* | -0.046*** | -0.024 | 0.147*** | 0.084*** |
| | (0.008) | (0.011) | (0.011) | (0.005) | (0.005) | (0.011) | (0.016) | (0.016) | (0.014) |
| Countries [*Ref: Mozambique*] | | | | | | | | | |
| Malawi | -0.083*** | 0.42*** | 0.188*** | 0.01 | -0.049*** | -0.215*** | -0.104*** | 0.235*** | -0.604*** |
| | (0.01) | (0.033) | (0.014) | (0.006) | (0.016) | (0.03) | (0.032) | (0.035) | (0.029) |
| Rwanda | -0.028*** | -0.066** | -0.014** | 0.015*** | -0.005 | 0.209*** | -0.093*** | 0.349*** | -0.612*** |
| | (0.004) | (0.031) | (0.007) | (0.004) | (0.015) | (0.028) | (0.027) | (0.031) | (0.025) |
| Uganda | -0.087*** | 0.482*** | -0.123*** | 0.032*** | -0.071*** | -0.147*** | -0.269*** | -0.033 | -0.665*** |
| | (0.021) | (0.044) | (0.028) | (0.012) | (0.022) | (0.046) | (0.049) | (0.05) | (0.045) |
| Zambia | -0.02*** | 0.031 | -0.012 | 0.036*** | 0.025 | 0.242*** | -0.242*** | 0.517*** | -0.536*** |
| | (0.005) | (0.032) | (0.01) | (0.005) | (0.017) | (0.029) | (0.03) | (0.032) | (0.027) |
| Observations | 19758 | 19758 | 19758 | 19758 | 19758 | 19758 | 19758 | 19758 | 19758 |
| R-squared | 0.011 | 0.197 | 0.041 | 0.009 | 0.037 | 0.098 | 0.066 | 0.108 | 0.056 |

Note: Standard errors in parentheses

*** p<0.01

** p<0.05

* p<0.1.

**Table 20. LPM regression for food groups consumed–leadership domain (Comfortable speaking in public in ≥ 1 context)—Mozambique.**

| | (1) | (2) | (3) | (4) | (5) | (6) | (7) | (8) | (9) |
|---|---|---|---|---|---|---|---|---|---|
| | Grain/ Roots | Legumes | Dairy | Organ meat | Eggs | Flesh protein | Vit A-rich dark-green leafy vegs | Other vit A-rich frts/ vegs | Other frts/ vegs |
| Public speaking | 0.018 | 0.024 | -0.006 | 0.003 | 0.027* | 0.053 | 0.051** | 0.095*** | 0.068*** |
| | (0.016) | (0.025) | (0.005) | (0.005) | (0.016) | (0.027) | (0.022) | (0.024) | (0.026) |
| SES index | -0.03 | -0.024 | 0.003 | 0.019 | 0.05 | -0.033 | 0.059* | 0.034 | -0.032 |
| | (0.04) | (0.036) | (0.008) | (0.013) | (0.033) | (0.046) | (0.032) | (0.033) | (0.039) |
| SES index squared | -0.025 | -0.123*** | 0.009** | 0.016 | 0.069** | 0.08** | 0.121*** | -0.019 | 0.036 |
| | (0.033) | (0.03) | (0.004) | (0.01) | (0.028) | (0.039) | (0.026) | (0.03) | (0.033) |
| Men's age | 0.002*** | 0.002 | 0.000 | 0.000 | 0.001 | 0.002 | -0.001 | 0.001 | 0.002 |
| | (0.001) | (0.001) | (0.000) | (0.000) | (0.001) | (0.001) | (0.001) | (0.001) | (0.001) |
| Women's age | -0.001 | -0.001 | 0.000*** | 0.000** | -0.001 | -0.002* | 0.000 | -0.004*** | -0.003** |
| | (0.001) | (0.001) | (0.000) | (0.000) | (0.001) | (0.001) | (0.001) | (0.001) | (0.001) |
| Women's education | 0.018*** | 0.007 | -0.001 | -0.001 | 0.014* | 0.004 | -0.01 | -0.009 | -0.006 |
| | (0.004) | (0.01) | (0.002) | (0.001) | (0.008) | (0.012) | (0.013) | (0.009) | (0.012) |
| Household size | 0.005** | 0.016*** | 0.001 | 0.000 | 0.001 | -0.002 | 0.01*** | 0.008* | 0.014*** |
| | (0.002) | (0.004) | (0.001) | (0.001) | (0.003) | (0.005) | (0.004) | (0.004) | (0.004) |
| Study location | 0.01*** | 0.001 | 0.000 | 0.001 | 0.006*** | 0.008*** | 0.000 | 0.019*** | 0.018*** |
| | (0.001) | (0.002) | (0.000) | (0.000) | (0.001) | (0.002) | (0.002) | (0.002) | (0.002) |
| Study month [*Ref: January*] | | | | | | | | | |
| February | -0.04*** | -0.78*** | 0.017*** | 0.011* | 0.196*** | 0.619*** | -0.106*** | 0.386*** | -0.368*** |
| | (0.011) | (0.024) | (0.005) | (0.006) | (0.021) | (0.028) | (0.019) | (0.034) | (0.029) |
| March | -0.006 | -0.851*** | 0.016*** | 0.023** | 0.135*** | 0.67*** | -0.239*** | 0.258*** | -0.466*** |
| | (0.02) | (0.033) | (0.005) | (0.009) | (0.023) | (0.04) | (0.034) | (0.039) | (0.039) |
| April | 0.081*** | -0.753*** | 0.026*** | 0.013* | 0.227*** | 0.667*** | -0.202*** | 0.443*** | -0.174*** |
| | (0.024) | (0.04) | (0.01) | (0.008) | (0.033) | (0.046) | (0.04) | (0.046) | (0.047) |
| November | 0.032** | -0.836*** | 0.021** | -0.001 | 0.073*** | 0.433*** | -0.022 | 0.64*** | -0.079* |
| | (0.013) | (0.042) | (0.009) | (0.005) | (0.025) | (0.051) | (0.036) | (0.052) | (0.042) |
| December | 0.014 | -0.599*** | 0.008** | 0.001 | 0.111*** | 0.228*** | -0.263*** | 0.185*** | -0.182*** |
| | (0.011) | (0.041) | (0.004) | (0.004) | (0.025) | (0.038) | (0.04) | (0.042) | (0.041) |
| Observations | 2604 | 2604 | 2604 | 2604 | 2604 | 2604 | 2604 | 2604 | 2604 |
| R-squared | 0.083 | 0.041 | 0.007 | 0.01 | 0.039 | 0.064 | 0.042 | 0.114 | 0.125 |

Note: Standard errors are in parentheses

*** p<0.01

** p<0.05

* p<0.1.

finding that when women were empowered in autonomy, and had input in production decisions, there was a shift from cereals cultivation towards fruits and vegetables. The reason is perhaps because foods with less economic value including fruits and vegetables are largely considered "women's crops" [71, 72].

There is limited literature that has found a positive association between empowerment in public speaking as an indicator of leadership empowerment and women's dietary diversity [45, 73]. A few studies have other measures of empowerment which subsume questions on public speaking. In northern Benin, Alaofè et al. [28] found that women's empowerment in leadership (which included questions on public speaking) was associated with improved women's dietary diversity. Similarly, Tsiboe et al. [22] using the WEAI measure in Ghana, found that

**Table 21. LPM regression for food groups consumed–leadership domain (Comfortable speaking in public in ≥ 1 context)—Rwanda.**

| | (1) | (2) | (3) | (4) | (5) | (6) | (7) | (8) | (9) |
|---|---|---|---|---|---|---|---|---|---|
| | Grain/ Roots | Legumes | Dairy | Organ meat | Eggs | Flesh protein | Vit A-rich dark-green leafy vegs | Other vit A-rich frts/ vegs | Other frts/ vegs |
| Public speaking | 0.059*** | 0.022 | 0.005 | 0.011* | -0.003 | 0.087 | 0.163*** | 0.067*** | 0.085*** |
| | (0.016) | (0.018) | (0.016) | (0.006) | (0.007) | (0.011) | (0.024) | (0.023) | (0.019) |
| SES index | 0.04 | -0.3** | 0.223* | 0.163** | 0.005 | 0.343** | 0.086 | -0.297* | 0.401*** |
| | (0.095) | (0.138) | (0.126) | (0.064) | (0.035) | (0.133) | (0.159) | (0.171) | (0.14) |
| SES index squared | 0.016 | -0.078* | 0.024 | 0.059*** | 0.000 | 0.101** | 0.024 | -0.085 | 0.143*** |
| | (0.032) | (0.046) | (0.042) | (0.021) | (0.012) | (0.044) | (0.055) | (0.058) | (0.048) |
| Men's age | -0.001 | 0.000 | 0.000 | 0.000 | 0.000 | 0.000 | 0.002** | 0.001 | 0.000 |
| | (0.000) | (0.001) | (0.000) | (0.000) | (0.000) | (0.000) | (0.001) | (0.001) | (0.001) |
| Women's age | 0.000 | 0.000 | -0.001 | 0.001** | -0.001*** | 0.001 | -0.003*** | -0.005*** | -0.002*** |
| | (0.000) | (0.001) | (0.001) | (0.000) | (0.000) | (0.001) | (0.001) | (0.001) | (0.001) |
| Women's education | 0.011*** | 0.004 | 0.022*** | 0.003 | 0.001 | 0.004 | 0.03*** | 0.029*** | 0.025*** |
| | (0.004) | (0.005) | (0.005) | (0.002) | (0.002) | (0.005) | (0.007) | (0.007) | (0.006) |
| Household size | 0.002 | 0.005** | 0.017*** | 0.000 | 0.002* | 0.006** | 0.005 | 0.009** | 0.008** |
| | (0.002) | (0.003) | (0.003) | (0.001) | (0.001) | (0.003) | (0.004) | (0.004) | (0.003) |
| Study location | 0.001** | 0.001* | -0.003*** | 0.000 | 0.000 | 0.003*** | 0.007*** | 0.005*** | 0.006*** |
| | (0.000) | (0.001) | (0.001) | (0.000) | (0.000) | (0.001) | (0.001) | (0.001) | (0.001) |
| Study month [*Ref: January*] | | | | | | | | | |
| December | 0.02** | -0.001 | 0.022** | 0.002 | 0.017*** | -0.031*** | 0.005 | 0.178*** | 0.107*** |
| | (0.009) | (0.012) | (0.011) | (0.005) | (0.004) | (0.01) | (0.016) | (0.016) | (0.014) |
| Observations | 4036 | 4036 | 4036 | 4036 | 4036 | 4036 | 4036 | 4036 | 4036 |
| R-squared | 0.01 | 0.009 | 0.045 | 0.006 | 0.012 | 0.022 | 0.036 | 0.05 | 0.036 |

Note: Standard errors are in parentheses

*** p<0.01

** p<0.05

* p<0.1.

disempowerment in leadership (which includes public speaking) was associated with reduced intake of proteins, carbohydrates, and fats.

Social norms regarding food taboos, myths, and perceptions might play an important role as to how women's empowerment affects their nutrition [41, 42, 74–76]. Using qualitative means, Puoane et al. [77] suggests that in SSA, in addition to nourishing the body, food is a sign of warmth, authority, acceptance and friendship where daily consumption of flesh proteins is associated with a high socioeconomic status while consumption of only vegetables is associated with a low socioeconomic status. This suggests that in households where social conflicts exist or where women are disempowered, food might be used as a tool of affection or correction. In rural South East Nigeria, Ekwochi et al. [40] suggests that women avoided the consumption of certain proteins and legumes due to perceived impact on child growth and development. In Rwanda, Imanizabayo [78] found that even where women understand the nutritional benefits of a balanced diet, dietary diversity is constrained by factors including lack of husband's support, time pressure and nature of daily jobs, and traditional and religious beliefs. In rural Mozambique, phrases like "Eat a Pig and Give Birth to a Pig" illustrate the effect of social norms on the consumption of certain food items for women of reproductive age [79, 80]. These social norms might have an important interaction, mediating, or

moderating effect on the women's empowerment-nutrition relationship, and should be an area for further research. As discussed, the diverse nature of the relationship between different empowerment indicators and women's dietary diversity and the consumption of specific food items suggests that different policies and interventions should be tailored towards improved consumption of specific food items. Policies that improve women's autonomy and input in agricultural production would improve women's consumption of food items in ways that might be different from those than policies that improve women's leadership opportunities. In addition, the varied associations across different countries further suggest that such policies should reflect contextual realities that influence women's empowerment and outcomes including food consumption and dietary diversity.

## Conclusion

In conclusion, the differential performance of the three indicators in the WEAI towards women's food consumption further suggests that different empowerment strategies might confer different benefits towards consumption of different food items and these benefits might vary across countries, but there are suggestions that some aspects of women's empowerment have a positive effect on women's nutrition. These variations in the empowerment-diet link also suggest that empowerment may give women greater autonomy in access to some of the more costly but nutrient-rich foods which are not typically produced on their own farm. The prevailing contextual factors within different countries including those that dictate women's roles and responsibilities are likely to play a significant role towards women's empowerment and dietary diversity.

## Supporting information

**S1 Table. Marginal effects of Poisson regression for WDDS–empowerment score.**
(DOCX)

**S2 Table. Marginal effects of Poisson regression for WDDS–prod domain (Aut in production).**
(DOCX)

**S3 Table. Marginal effects of Poisson regression for WDDS–prod domain (Input in productive decisions).**
(DOCX)

**S4 Table. Marginal effects of Poisson regression results for WDDS–resources domains (asset ownership).**
(DOCX)

**S5 Table. Marginal effects of Poisson regression results for WDDS–resources domain (asset sale, purchase, and transfer).**
(DOCX)

**S6 Table. Marginal effects of Poisson regression results for WDDS–resources domain (input in $\geq 1$ credit source).**
(DOCX)

**S7 Table. Marginal effects of Poisson regression results for WDDS–income domain (Input in at least 1 income decision).**
(DOCX)

**S8 Table. Marginal effects of Poisson regression results for WDDS–leadership domain (group membership).**
(DOCX)

**S9 Table. Marginal effects of Poisson for WDDS–leadership domain (comfortable speaking in public in $\geq$1 context).**
(DOCX)

**S10 Table. Marginal effects of Poisson for WDDS–time domain (Non-excessive workload [$<$10.5hrs in 24hrs]).**
(DOCX)

**S11 Table. Marginal effects of Poisson regression for WDDS–time domain (satisfaction with leisure time).**
(DOCX)

**S12 Table. Marginal effects of Logistic regression for food groups consumed–production domain (autonomy in $\geq$ 1 activity linked to production)–pooled.**
(DOCX)

**S13 Table. Marginal effects of Logistic regression for food groups consumed–production domain (Input in $\geq$ 2 productive decisions)–pooled.**
(DOCX)

**S14 Table. Marginal effects of Logistic regression for food groups consumed–production domain (Input in $\geq$ 2 productive decisions)–Rwanda.**
(DOCX)

**S15 Table. Marginal effects of Logistic regression for food groups consumed–Leadership domain (Comfortable speaking in public in $\geq$ 1 context)–pooled.**
(DOCX)

**S16 Table. Marginal effects of Logistic regression for food groups consumed–Leadership domain (Comfortable speaking in public in $\geq$ 1 context)–Mozambique.**
(DOCX)

**S17 Table. Marginal effects of Logistic regression for food groups consumed–Leadership domain (Comfortable speaking in public in $\geq$ 1 context)–Rwanda.**
(DOCX)

**S18 Table. Correlation matrix–pooled data.**
(DOCX)

## Acknowledgments

The authors would like to thank Craig Janes and Suneetha Kadiyala for their critical comments on the paper as the lead author's PhD thesis committee member and external examiner, respectively.

## Author Contributions

**Conceptualization:** Michael Nnachebe Onah, Sue Horton.

**Data curation:** Michael Nnachebe Onah.

**Formal analysis:** Michael Nnachebe Onah.

**Funding acquisition:** Michael Nnachebe Onah.

**Investigation:** Michael Nnachebe Onah.

**Methodology:** Michael Nnachebe Onah, John Hoddinott.

**Software:** Michael Nnachebe Onah, John Hoddinott.

**Supervision:** Sue Horton, John Hoddinott.

**Writing – original draft:** Michael Nnachebe Onah, Sue Horton, John Hoddinott.

**Writing – review & editing:** Michael Nnachebe Onah, Sue Horton, John Hoddinott.

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
