## [Decision Letter · Decision Letter 0]

11 Sep 2020

PONE-D-20-24920

What empowerment indicators are important for food consumption for women? Evidence from 5 sub-Sahara African countries

PLOS ONE

Dear Dr. Onah,

Thank you for submitting your manuscript to PLOS ONE. After careful consideration, we feel that it has merit but does not fully meet PLOS ONE’s publication criteria as it currently stands. Therefore, we invite you to submit a revised version of the manuscript that addresses the points raised during the review process.

We look forward to receiving your revised manuscript.

Kind regards,

Yacob Zereyesus, Ph.D.

Academic Editor

PLOS ONE

Journal Requirements:

"No funding statement "

"No conflict of interest "

Reviewers' comments:

Reviewer's Responses to Questions

**Comments to the Author**

1. Is the manuscript technically sound, and do the data support the conclusions?

Reviewer #1: Partly

2. Has the statistical analysis been performed appropriately and rigorously? 

Reviewer #1: No

3. Have the authors made all data underlying the findings in their manuscript fully available?

Reviewer #1: Yes

4. Is the manuscript presented in an intelligible fashion and written in standard English?

Reviewer #1: Yes

5. Review Comments to the Author

Reviewer #1: Comments:

The paper covers an interesting area, and it is well-written. My comments below are intended to improve the paper.

1. It would be helpful to have a discussion about social norms and nutritional status in each of the five countries in the study, especially since the study recommends to tailor empowerment strategies around contextual factors. What are the contextual factors around gender norms and nutrition in each country?

2. There is now a body of literature that links women’s empowerment to women’s nutritional status. I recommend to have a summary of the literature on what they say in the introduction, highlighting which aspects of empowerment have been associated with improved nutrition. That should set up the paper in stating how the paper motivates and contributes to existing literature.

3. The regression results (marginal effects from the Poisson regression) are not in the body of the paper, but are attached as supplemental material. I recommend that these results be included in the body of the paper because these results should form the core of the paper.

4. The household hunger score is likely an outcome variable and not an independent variable for women’s dietary diversity. The argument to include the household hunger score as an independent variable as it is written in the paper is not convincing as it is likely endogenous.

5. The variables “Study location” and “Study month” are included in the regressions as continuous variables in pooled and country level regressions. Usually, these variables are included in the regressions as dummy variables because, for example, study locations do not affect the outcome variable in a continuous manner.

6. For the pooled regressions, I recommend that dummy variables for each country be included because they are likely to have country level differences that affect the outcome variable. However, I am personally not in favor of pooled regressions with different countries because they assume that the independent variables have the same effects on the outcome variable across five countries. That is a very strong assumption to make.

7. The asset index has a negative relationship with women’s dietary diversity in Malawi, Zambia, and Mozambique (though not significant for the latter two). I would check what variables are contained in the asset index to make sure that they are positively correlated with each other. There might be some variables that have negative relationship with socio-economic status. Also, please consider using a nonlinear approach for the asset index by including the index and index squared.

8. There are some variables, such as number of children or land size, that could affect WDD. I would consider adding some socio-demographic variables in the regressions.

9. The workload variable in the WEAI is complicated because the time spent in at three broad areas conflict with each other, and with women’s empowerment. The variable includes time spent in domestic work, which relates to access to water and household fuel; agricultural work, which also relates to access to agricultural technology; and paid work. Spending time in paid work could actually increase women’s empowerment and WDD because they earn income. Therefore, being overworked could increase women’s empowerment. Time spent in agricultural work could increase income and empowerment, or it could lead to disempowerment. It might be worth breaking the variable down to look at which aspects of time spent affect WDD.

6. PLOS authors have the option to publish the peer review history of their article (what does this mean?). If published, this will include your full peer review and any attached files.

Reviewer #1: No

---

## [Author Response · Author response to Decision Letter 0]

18 Nov 2020

Thank you for the careful appraisal and detailed comments from both editor and reviewers. We value the opportunity to revise and improve this paper. Find below our point by point response to reviewer comments. Where appropriate, changes have been made in track changes. 

Reviewer #1: Comments:

The paper covers an interesting area, and it is well-written. My comments below are intended to improve the paper.

1. It would be helpful to have a discussion about social norms and nutritional status in each of the five countries in the study, especially since the study recommends to tailor empowerment strategies around contextual factors. What are the contextual factors around gender norms and nutrition in each country?

Response: Many thanks for this suggestion. We have included a brief summary discussion of the relationship between social norms and contextual factors and women’s nutrition. 

2. There is now a body of literature that links women’s empowerment to women’s nutritional status. I recommend to have a summary of the literature on what they say in the introduction, highlighting which aspects of empowerment have been associated with improved nutrition. That should set up the paper in stating how the paper motivates and contributes to existing literature.

Response: We have followed this suggestion and have now included a paragraph on the links between women’s empowerment and women’s nutrition. 

3. The regression results (marginal effects from the Poisson regression) are not in the body of the paper, but are attached as supplemental material. I recommend that these results be included in the body of the paper because these results should form the core of the paper.

Response: We thank you for this suggestion. The Poisson and logistic regression analyses were performed as confirmatory tests of the OLS and LPM models and produced similar results. The OLS and LPMs are reported in the manuscript and we stated that the Poisson and logistic regressions confirm the study findings and placed the outputs in the supplementary files. If the referee feels strongly about this point, we could include them in the main part of the paper but our preference – in the interests of keeping the paper short – is to leave them as part of the supplementary material. We have also replaced the graphical illustration of the OLS and LPM results with Tables for easier interpretation. 

4. The household hunger score is likely an outcome variable and not an independent variable for women’s dietary diversity. The argument to include the household hunger score as an independent variable as it is written in the paper is not convincing as it is likely endogenous.

Response: You are correct. We performed an endogeneity test (Durbin–Wu–Hausman test) and indeed hunger scale is endogenous hence we excluded it from the updated analyses. 

5. The variables “Study location” and “Study month” are included in the regressions as continuous variables in pooled and country level regressions. Usually, these variables are included in the regressions as dummy variables because, for example, study locations do not affect the outcome variable in a continuous manner.

Response: We have updated the regression analyses by treating study month as a dummy variable. However, for illustrative purposes, study location was presented as continuous since there were about 38 study locations in the pooled data (Mozambique; 22, Malawi; 7, Rwanda; 27, Uganda; 37, and Zambia; 6). In addition, since the focus of the study was not the interpretation of the effect of covariates, we focused on the interpretation of core independent variables. 

6. For the pooled regressions, I recommend that dummy variables for each country be included because they are likely to have country level differences that affect the outcome variable. However, I am personally not in favor of pooled regressions with different countries because they assume that the independent variables have the same effects on the outcome variable across five countries. That is a very strong assumption to make.

Response: We have included a dummy variable for each country in the updated pooled analyses 

7. The asset index has a negative relationship with women’s dietary diversity in Malawi, Zambia, and Mozambique (though not significant for the latter two). I would check what variables are contained in the asset index to make sure that they are positively correlated with each other. There might be some variables that have negative relationship with socio-economic status. Also, please consider using a nonlinear approach for the asset index by including the index and index squared.

Response: We performed a correlation analyses on the pooled data (Suppl Table 21) and retained only variables (listed in methods section) with moderate to high correlation in the principal component analyses. Indeed, in some countries, the asset index score is negatively associated with WDDS. In such cases, asset index appears to be more positively and significantly associated with the consumption of different food groups. This suggests that perhaps SES index is a better predictor of the consumption of different food groups and not WDDS. The consumption of some of these food groups are also associated with higher SES in literature. We also used a nonlinear approach by developing and including an asset index squared variable in the models. Further, we did not focus on the interpretation of the association coefficients since this was not the focus of the present study.

8. There are some variables, such as number of children or land size, that could affect WDD. I would consider adding some socio-demographic variables in the regressions.

Response: We agree that there are important but missing control variables that are potentially associated with WDDS. However, we included all available control variables in the data. We have also mentioned this as a study limitation. 

9. The workload variable in the WEAI is complicated because the time spent in at three broad areas conflict with each other, and with women’s empowerment. The variable includes time spent in domestic work, which relates to access to water and household fuel; agricultural work, which also relates to access to agricultural technology; and paid work. Spending time in paid work could actually increase women’s empowerment and WDD because they earn income. Therefore, being overworked could increase women’s empowerment. Time spent in agricultural work could increase income and empowerment, or it could lead to disempowerment. It might be worth breaking the variable down to look at which aspects of time spent affect WDD.

Response: We agree that the type of work matter however, paid work may not always increase empowerment. For instance, different paid work like exhausting paid work (e.g. weeding other people’s farmers) might have different effects than white collar work. The variable definitions in the baseline data are not ideal to do this more detailed analysis. We have also removed the discussion of workload from the manuscript since the association with WDDS is not significant in the updated analyses.

---

## [Decision Letter · Decision Letter 1]

8 Feb 2021

PONE-D-20-24920R1

What empowerment indicators are important for food consumption for women? Evidence from 5 sub-Sahara African countries

PLOS ONE

Dear Dr. Onah,

Thank you for submitting your manuscript to PLOS ONE. After careful consideration, we feel that it has merit but does not fully meet PLOS ONE’s publication criteria as it currently stands. Therefore, we invite you to submit a revised version of the manuscript that addresses the points raised during the review process.

We look forward to receiving your revised manuscript.

Kind regards,

Kannan Navaneetham, PhD

Academic Editor

PLOS ONE

Reviewers' comments:

Reviewer's Responses to Questions

**Comments to the Author**

1. If the authors have adequately addressed your comments raised in a previous round of review and you feel that this manuscript is now acceptable for publication, you may indicate that here to bypass the “Comments to the Author” section, enter your conflict of interest statement in the “Confidential to Editor” section, and submit your "Accept" recommendation.

Reviewer #1: (No Response)

2. Is the manuscript technically sound, and do the data support the conclusions?

Reviewer #1: Yes

3. Has the statistical analysis been performed appropriately and rigorously? 

Reviewer #1: Yes

4. Have the authors made all data underlying the findings in their manuscript fully available?

Reviewer #1: Yes

5. Is the manuscript presented in an intelligible fashion and written in standard English?

Reviewer #1: Yes

6. Review Comments to the Author

Reviewer #1: Thank you for addressing the comments on empirical strategy. I do not have further comments on this aspect of the paper.

However, I feel that my comments 1 (norms and country context on nutrition) and 2 (literature on empowerment and women's nutrition) have not been adequately addressed. The motivation of the paper and the literature on the pathway of linking empowerment to nutrition are missing, so the paper starts with the introduction and goes next to data description. The motivation, the nutrition framework, and how the paper contributes to the literature need greater detail. How different is this paper from the other papers on this topic? The discussion of results and conclusion are also quite generic. I think adding these sections and giving more detail would provide a stronger case for why this paper should be important.

7. PLOS authors have the option to publish the peer review history of their article (what does this mean?). If published, this will include your full peer review and any attached files.

Reviewer #1: No

---

## [Author Response · Author response to Decision Letter 1]

24 Mar 2021

Dear Editor,

Revisions for manuscript ref: PONE-D-20-24920 “What empowerment indicators are important for food consumption for women? Evidence from 5 sub-Sahara African countries”

Thank you for the careful appraisal and detailed comments from both editor and reviewers. We value the opportunity to revise and improve this paper. Find below our point by point response to reviewer comments. Where appropriate, changes have been made in track changes. 

Reviewer #1: Thank you for addressing the comments on empirical strategy. I do not have further comments on this aspect of the paper.

However, I feel that my comments 1 (norms and country context on nutrition) and 2 (literature on empowerment and women's nutrition) have not been adequately addressed. The motivation of the paper and the literature on the pathway of linking empowerment to nutrition are missing, so the paper starts with the introduction and goes next to data description. The motivation, the nutrition framework, and how the paper contributes to the literature need greater detail. How different is this paper from the other papers on this topic? The discussion of results and conclusion are also quite generic. I think adding these sections and giving more detail would provide a stronger case for why this paper should be important.

Response: Many thanks for your comments. We have updated the introduction and discussion sections to include more details on the link between context and nutrition and have also made a better motivation for the study.

---

## [Editor Report · Decision Letter 2]

30 Mar 2021

What empowerment indicators are important for food consumption for women? Evidence from 5 sub-Sahara African countries

PONE-D-20-24920R2

Dear Dr. Onah,

We’re pleased to inform you that your manuscript has been judged scientifically suitable for publication and will be formally accepted for publication once it meets all outstanding technical requirements.

Kind regards,

Kannan Navaneetham, PhD

Academic Editor

PLOS ONE
---

## [Editor Report · Acceptance letter]

5 Apr 2021

PONE-D-20-24920R2 

What empowerment indicators are important for food consumption for women? Evidence from 5 sub-Sahara African countries 

Dear Dr. Onah:

I'm pleased to inform you that your manuscript has been deemed suitable for publication in PLOS ONE. Congratulations! Your manuscript is now with our production department. 

Kind regards, 

on behalf of

Prof. Kannan Navaneetham 

Academic Editor

PLOS ONE